# Search for Majorana neutrinos exploiting millikelvin cryogenics with CUORE

The CUORE Collaboration*

The possibility that neutrinos may be their own antiparticles, unique among the known fundamental particles, arises from the symmetric theory of fermions proposed by Ettore Majorana in 1937[1]. Given the profound consequences of such Majorana neutrinos, among which is a potential explanation for the matter–antimatter asymmetry of the universe via leptogenesis[2], the Majorana nature of neutrinos commands intense experimental scrutiny globally; one of the primary experimental probes is neutrinoless double beta ($0\nu\beta\beta$) decay. Here we show results from the search for $0\nu\beta\beta$ decay of $^{130}$Te, using the latest advanced cryogenic calorimeters with the CUORE experiment[3]. CUORE, operating just 10 millikelvin above absolute zero, has pushed the state of the art on three frontiers: the sheer mass held at such ultralow temperatures, operational longevity, and the low levels of ionizing radiation emanating from the cryogenic infrastructure. We find no evidence for $0\nu\beta\beta$ decay and set a lower bound of the process half-life as $2.2 \times 10^{25}$ years at a 90 per cent credibility interval. We discuss potential applications of the advances made with CUORE to other fields such as direct dark matter, neutrino and nuclear physics searches and large-scale quantum computing, which can benefit from sustained operation of large payloads in a low-radioactivity, ultralow-temperature cryogenic environment.

The standard model of particle physics is a successful paradigm for the number, properties and interactions of fundamental particles. Nevertheless, the observation of neutrino oscillations indicates the incompleteness of the standard model: they imply non-vanishing neutrino masses, requiring an extension of the standard model, and violate three accidental symmetries connected to the flavour lepton numbers $L_e$, $L_\mu$ and $L_\tau$, leaving the difference between the baryon and lepton number, $B - L$, as the only unprobed quantity. A promising process to experimentally test $B - L$ is neutrinoless double beta ($0\nu\beta\beta$) decay, in which a nucleus of mass number $A$ and charge $Z$ decays by the emission of only two electrons: $(A, Z) \rightarrow (A, Z + 2) + 2e^-$. We highlight that this process creates two electrons, namely two matter particles[4]. This decay can be mediated by various non-standard model mechanisms involving Majorana neutrino masses. A minimal extension of the standard model Lagrangian adds heavy Majorana neutrinos that mix with the known neutrinos to produce a set of light Majorana neutrinos, explaining the observed light neutrino masses[5] and at the same time providing a mechanism to explain the baryon asymmetry in the universe[2]. At this time, experimental searches for $0\nu\beta\beta$ decay are the most sensitive means to corroborate this framework.

The $0\nu\beta\beta$ decay signature is a peak in the spectrum of summed energy of the two emitted electrons at the mass difference ($Q_{\beta\beta}$) between the parent and daughter nuclei. A worldwide quest is ongoing, involving a range of nuclei such as $^{76}$Ge[6,7], $^{136}$Xe[8,9] and $^{130}$Te. The latter, in the form of TeO$_2$ cryogenic calorimeters, is used by the Cryogenic Underground Observatory for Rare Events, CUORE[10,11].

To fully exploit the potential of TeO$_2$ crystals as cryogenic calorimeters, the CUORE Collaboration designed and built to our knowledge the largest dilution refrigerator ever constructed, capable of cooling approximately 1.5 t of material to a temperature of approximately 10 mK and maintaining it for years with a 90% duty cycle (1 t = 1,000 kg). In this Article, we describe the performance of CUORE over a four-year measurement campaign and the results of a new high-sensitivity $0\nu\beta\beta$ decay search with over 1 t yr of TeO$_2$ exposure.

## The CUORE experiment

CUORE is the culmination of thirty years of $0\nu\beta\beta$ decay searches with TeO$_2$ cryogenic calorimeters[12]. $^{130}$Te benefits both from a high natural isotopic abundance of approximately 34%[13] and a high $Q_{\beta\beta}$ of 2,527.5 keV[14], placing the $0\nu\beta\beta$ decay region of interest above most natural $\gamma$-emitting radioactive backgrounds. The detector is an array of 988 $^{nat}$TeO$_2$ cubic crystals[15] (Fig. 1) of $5 \times 5 \times 5$ cm$^3$ size and ~750 g mass, for a total mass of 742 kg, which corresponds to 206 kg of $^{130}$Te. The array is arranged as 19 towers, each comprised of 13 floors containing four crystals. The crystals are operated as cryogenic calorimeters[16] at a temperature of approximately 10 mK. To achieve this low-temperature environment, a novel cryogenic infrastructure—the CUORE cryostat—has been realized.

In a cryogenic calorimeter, the energy deposited by impinging radiation in the absorber crystal is turned into heat, resulting in a temperature rise (Extended Data Fig. 1). Each CUORE crystal (Fig. 1c) is instrumented with a neutron-transmutation-doped germanium thermistor (NTD)[17] that converts thermal pulses into electric signals and a heater[18] to inject reference heat pulses for thermal gain stabilization[19]. Thermal signals can be induced by electrons emitted in $0\nu\beta\beta$ decays but

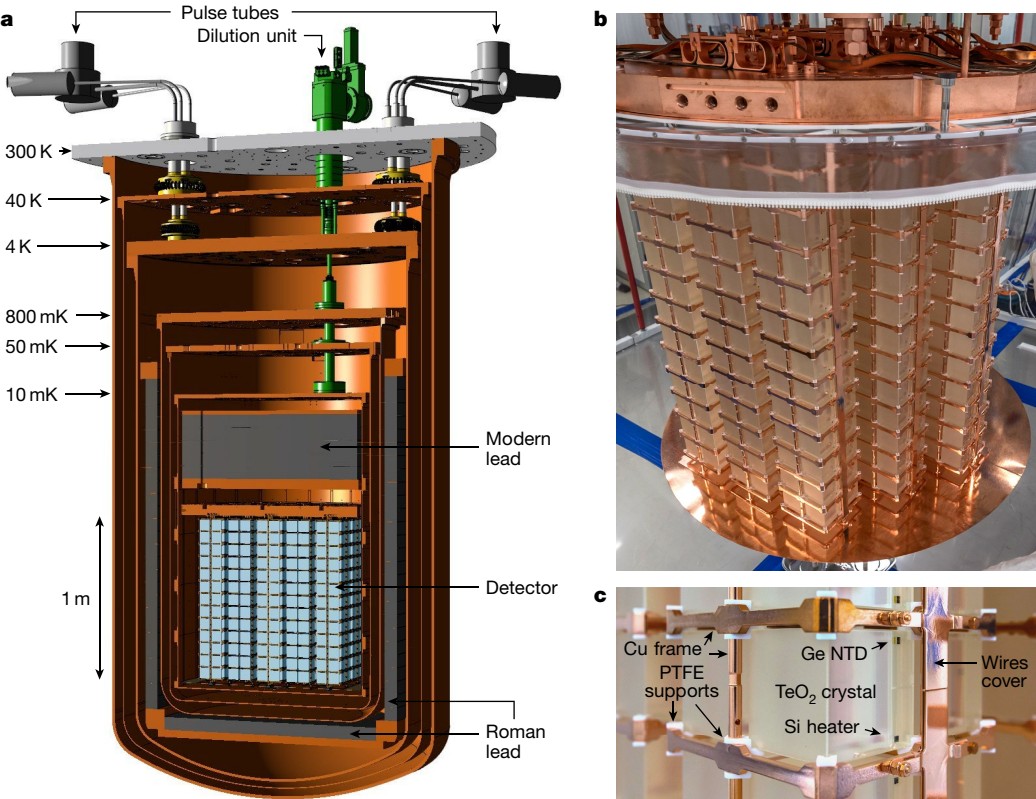

**Fig. 1 | The CUORE detector. a**, Rendering of the six-stage cryostat, with the pulse tubes and dilution unit, the internal low-radioactivity modern and Roman lead shields, and the array of 988 TeO$_2$ crystals (light blue). **b**, The detector after installation. The plastic ring was used during assembly for radon protection. **c**, One of the calorimeters instrumented with an NTD Ge thermistor which measures the temperature increase induced by absorbed radiation. The Si heater is used to inject pulses for thermal gain stabilization. The polytetrafluoroethylene (PTFE) supports and the gold wires instrumenting the NTD and the heater provide the thermal link between the crystal and the heat bath, that is, the Cu frames[24].

also other background radiation, for example, $\gamma$ and $\alpha$ particles from residual radioactive contaminants or cosmic ray muons.

CUORE is protected by several means against backgrounds that can mimic a $0\nu\beta\beta$ decay. It is located underground at the Laboratori Nazionali del Gran Sasso (LNGS) of INFN, Italy, under a rock overburden equivalent to approximately 3,600 m of water, which shields from hadronic cosmic rays and reduces the muon flux by six orders of magnitude. Environmental $\gamma$ backgrounds are suppressed by a 30-cm layer of low-radioactivity lead above the detector (Fig. 1), a 6-cm-thick lateral and bottom shield of $^{210}$Pb-depleted ancient lead recovered from a Roman shipwreck[20] (Extended Data Fig. 2), and a 25-cm-thick lead shield outside the cryostat. Environmental neutrons are suppressed by a 20-cm layer of polyethylene and a thin layer of boric acid outside the external lead shield. Finally, radioactive contaminants in the crystals and in the adjacent structures are minimized by careful screening of material for radio-purity and use of high-efficiency cleaning procedures and manipulation protocols[21].

## Cryogenic innovation and performance

Dilution refrigerator technology was originally proposed in the 1950s[22] and underwent considerable development in the 1980s driven also by the application of cryogenic calorimeters for single-particle detection[23]. Gradually, experimental volumes of the order of tens of litres capable of hosting cold masses of up to 60 kg at 10 mK temperature[24] were achieved. Ultimately, detectors were limited by the capacity, duty cycle and radio-purity of commercial or near-commercial cryogenic systems. In the context of this history, the CUORE cryostat represents a major advance in cryogenic technology, reaching an experimental volume of approximately 1 m$^3$ and a cold mass of 1.5 t (detectors,

holders, shields) at 10 mK, which corresponds to a 20-fold improvement in experimental volume and target mass compared to the previous state of the art at this temperature scale. Prior to CUORE, the ultimate temperature for comparable target masses was in the resonant-mass gravitational antenna community at 65 mK[23].

The CUORE detector is hosted in a multistage cryogen-free cryostat[25] (Fig. 1), equipped with five pulse tube cryocoolers that avoid pre-cooling with a liquid helium bath, thus enabling a high duty cycle. A custom-designed dilution unit with a double condensing line for redundancy provides more than 4 μW cooling power at 10 mK. The cryostat is uniquely designed to provide the necessary i) cooling power and temperature stability over a time scale of years, ii) very low radioactivity environment, and iii) extremely low-vibration conditions. As shown in Fig. 2a, b, CUORE became operational in 2017, with the initial period mostly devoted to characterization and optimization campaigns. Since 2019, the data-taking has proceeded smoothly with a duty cycle of approximately 90%. Figure 2d shows that the temperature stability achieved is at the level of 0.2% (±3$\sigma$ range) over a period in excess of one year. Such a stability is important to achieve a uniform response of all detectors over time. The CUORE calorimeters are sensitive to thermal signals and feature an intrinsic thermal fluctuation limit of approximately 0.5 keV, so any process inducing heat dissipation equal to or greater than 0.5 keV degrades the energy resolution. Mechanical vibrations can be transferred to the inner components and produce heat through friction. To minimize the impact of vibrational noise, the calorimeter array is mechanically decoupled from the cryostat by a custom suspension system. Vibrations induced by the pulse tubes at the 1.4-Hz operational frequency and its harmonics are particularly relevant. In CUORE, we actively tune the pulse tube relative phases for

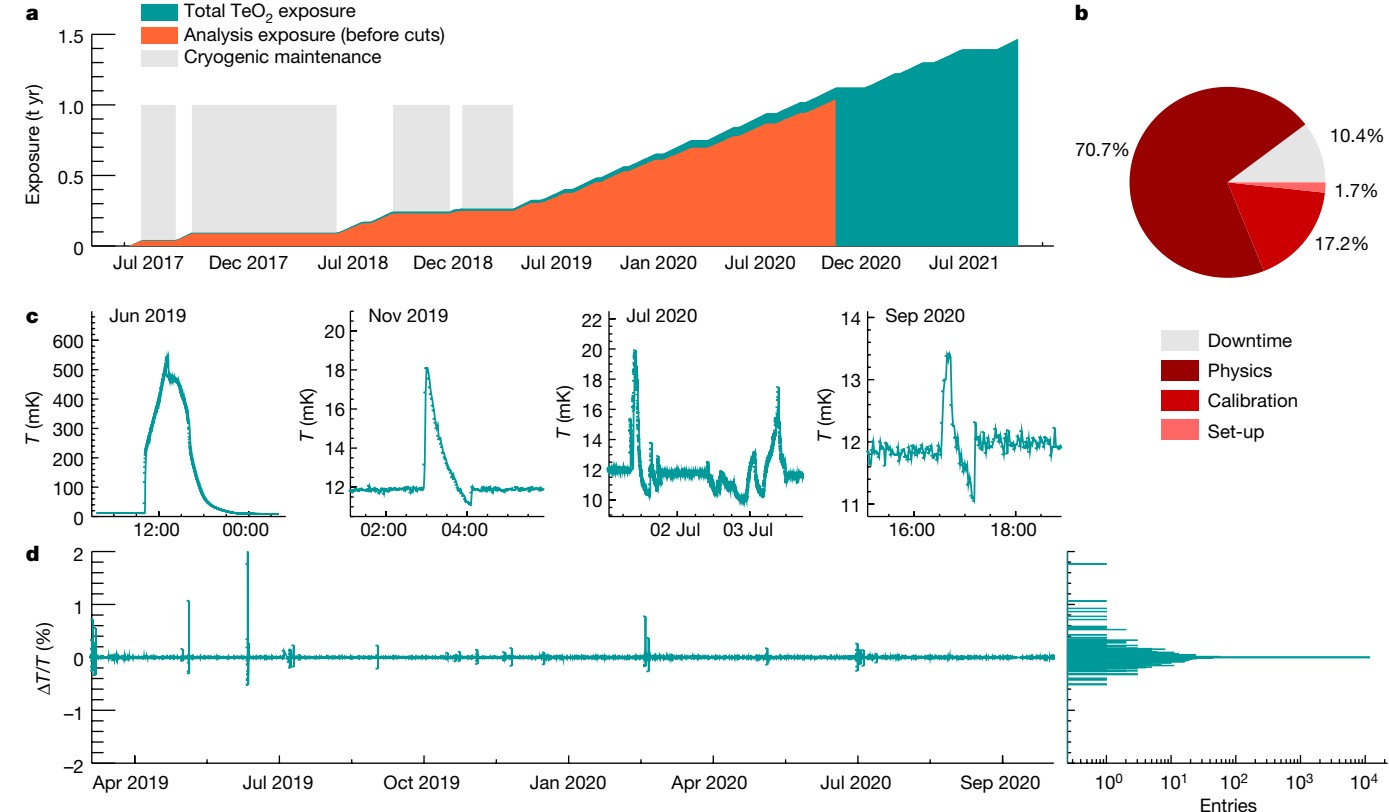

**Fig. 2 | Cryogenic performance. a**, The exposure accumulated by CUORE (teal), along with the exposure used for this analysis (orange). Parts of 2017 and 2018 were dedicated to maintenance and optimization of the cryogenic set-up. **b**, Since then, CUORE has been operating stably with a 90% duty cycle (March 2019–October 2020). **c**, Examples of temperature instabilities induced by external causes. From left to right: blackout (June 2019), earthquake in Albania of magnitude 6.4, 520 km away (November 2019), regular maintenance (July 2020), and insertion of calibration sources (September 2020). **d**, The temperature stability of CUORE over ~1 yr of continuous operation, shown by a plot of relative temperature fluctuation versus time, and a histogram of the same data. (1 t yr = 1,000 kg yr.).

vibration cancellation[26] (Fig. 3). This solution is transferable to any cryogenic application involving signals in the same bandwidth of the pulse-tube-induced noise.

CUORE now collects sensitive exposure with 984 out of 988 calorimeters, at a rate that is, to our knowledge, unprecedented for cryogenic particle detectors. Below, we describe the data treatment and $0\nu\beta\beta$ decay search with greater than 1 t yr of TeO$_2$ exposure.

## Data analysis and results

CUORE data are subdivided into datasets of 1–2 months of physics data, separated by a few days of calibration data collected with the detector exposed to $^{232}$Th and/or $^{60}$Co sources.

The voltage across each NTD is amplified, passed through an anti-aliasing filter, and continuously digitized with a 1-kHz sampling frequency[27,28]. We identify thermal pulses in the data stream and compute the pulse amplitudes, applying optimum filters that maximize the frequency-dependent signal-to-noise ratio[29]. To monitor and correct for possible drifts of the thermal gain of the detectors we exploit two 'standard candles': monoenergetic heater pulses for the calorimeters with functioning and stable heaters (>95% of the total), and events from the 2,615-keV $^{208}$Tl calibration line for the remainder. Drift-stabilized pulse amplitudes are converted to energy using the regularly acquired source calibration data[30]. We blind the $0\nu\beta\beta$ search via a data-salting procedure that produces an artificial peak at $Q_{\beta\beta}$[30]. Once the full analysis procedure is finalized and frozen, we reverse the salting.

To simplify the analysis, we eliminate data from periods affected by high noise or failed data processing, which amounts to 5% of the

exposure. Furthermore, calorimeters with greater than 19-keV full width at half maximum (FWHM) energy resolution at the 2,615-keV calibration line are discarded, adding 3% loss in exposure. In addition to these so-called base cuts, the following second-level cuts are then applied to suppress single background-like or low-quality events. Monte Carlo simulations show that approximately 88% of $0\nu\beta\beta$ decay events release their full energy in a single crystal[31]. Hence, we apply an anti-coincidence cut that excludes events depositing energy in more than one crystal. Finally, we use pulse shape discrimination to eliminate pulses that are consistent with more than one energy deposit in

**Table 1 | Main parameters for the $0\nu\beta\beta$ analysis**

| Parameter | Value |
|---|---|
| Number of datasets | 15 |
| TeO$_2$ exposure | 1,038.4 kg yr |
| $^{130}$Te exposure | 288.8 kg yr |
| FWHM at 2,615 keV in calibration data | 7.78(3) keV |
| FWHM at $Q_{\beta\beta}$ in physics data | 7.8(5) keV |
| Total analysis efficiency (data) | 92.4(2)% |
| Reconstruction efficiency | 96.418(2)% |
| Anticoincidence efficiency | 99.3(1)% |
| PSD efficiency | 96.4(2)% |
| Containment efficiency (Monte Carlo) | 88.35(9)%[30] |

The resolution and efficiencies are exposure-weighted average values.

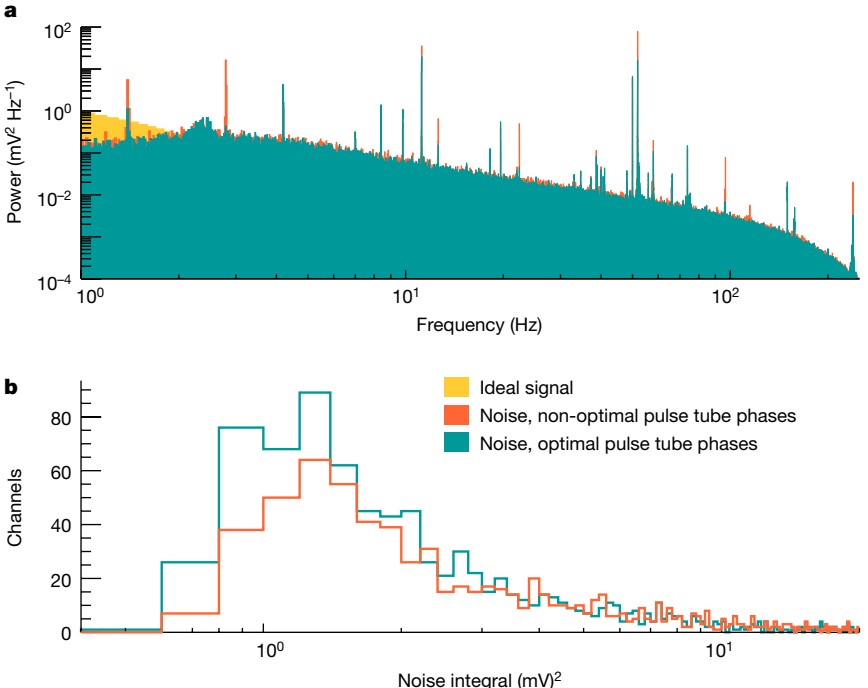

**Fig. 3 | Pulse tube phase optimization. a**, Frequency spectrum of the noise for a random combination of the pulse tube phases (orange) and after the active phase tuning (teal). For reference, the frequency spectrum of physical signals is also reported. **b**, Integral of the power spectrum at the pulse tube frequency (1.4 Hz) and its harmonics before and after active noise cancellation.

the pulse time window, pulses with a non-physical shape, and excessively noisy pulses that survived the previous selections (Extended Data Fig. 3). The selection efficiencies are summarized in Table 1, with details provided in Methods.

The detector response to a monoenergetic energy deposition is an important input to the $0\nu\beta\beta$ decay search. We empirically model the response function of each calorimeter as a sum of three equal-width Gaussians and determine the function parameters from a fit to the 2,615-keV calibration line[3]. As a characteristic indicator of the overall energy resolution of the calorimeters we quote the exposure-weighted harmonic mean FWHM of the detectors at this calibration line, 7.78 ± 0.03 keV. All values are reported as mean ± s.d.

We quantify the scaling of energy resolution with energy and investigate energy reconstruction bias—that is, the deviation of reconstructed $\gamma$-line positions from the literature values—by fitting the calorimeter response functions to prominent $\gamma$ lines in the physics data, allowing the peak means and widths to vary in the fit. The bias is allowed to scale as a quadratic function of energy, as done in our previous result[32], whereas the resolution scaling has been changed to a linear function of energy, following studies showing that it was overparameterized by a quadratic scaling. The results, extrapolated to $Q_{\beta\beta}$, are an exposure-weighted harmonic mean FWHM energy resolution of 7.8 ± 0.5 keV and an energy bias of less than 0.7 keV. We summarize all the relevant analysis quantities in Table 1.

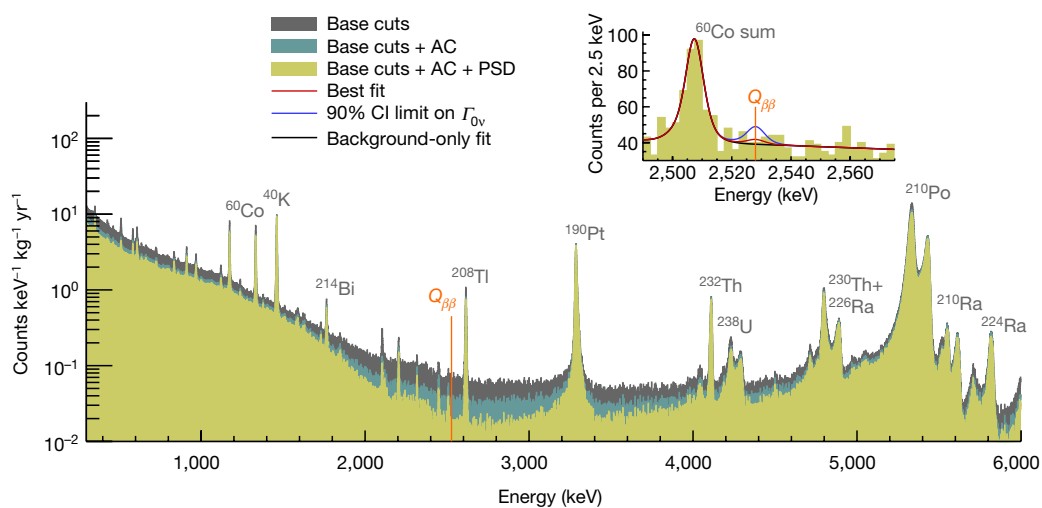

**Fig. 4 | Physics spectrum for 1,038.4 kg yr of TeO₂ exposure.** We separately show the effects of the base cuts, the anti-coincidence (AC) cut, and the pulse shape discrimination (PSD). The most prominent background peaks in the spectrum are highlighted. Inset, the region of interest after all selection cuts, with the best-fit curve (solid red), the best-fit curve with the $0\nu\beta\beta$ rate fixed to the 90% CI limit (blue), and background-only fit (black) superimposed.

Figure 4 shows the full energy spectrum along with the [2,490, 2,575] keV fit region, which contains only one background peak at 2,505.7 keV from the simultaneous absorption of two coincident $\gamma$ rays from $^{60}$Co in the same crystal. We estimate that around 90% of the continuum background consists of degraded $\alpha$ particles from radioactive contaminants of the support structure surface, and the other approximately 10% are multi-Compton scattered 2,615-keV $\gamma$ events[31,33].

We run an unbinned Bayesian fit with uniform non-negative priors on the background and 0$\nu\beta\beta$ decay rates. The fit with a background-only model—that is, excluding the 0$\nu\beta\beta$ component—yields a mean background rate of $(1.49 \pm 0.04) \times 10^{-2}$ counts keV$^{-1}$ kg$^{-1}$ yr$^{-1}$ at $Q_{\beta\beta}$ for a corresponding median exclusion sensitivity of $T_{1/2}^{0\nu} > 2.8 \times 10^{25}$ yr (90% credibility interval (CI)). The fit with the signal-plus-background model shows no evidence for 0$\nu\beta\beta$ decay. We find the best fit at $\Gamma_{0\nu} = (0.9 \pm 1.4) \times 10^{-26}$ yr$^{-1}$ and set a limit on the process half-life of $T_{1/2}^{0\nu} > 2.2 \times 10^{25}$ yr (90% CI). Systematic uncertainties are included as nuisance parameters and affect both the best fit and the limit by 0.8% (Extended Data Table 1). Compared to the sensitivity, the probability of getting a stronger limit is 72%. This represents, to our knowledge, the current world-leading 0$\nu\beta\beta$ sensitivity for $^{130}$Te, having improved in accordance with our increased exposure since our previous result[32], and although the actual limit is weaker, it is well within the range of possible outcomes due to statistical fluctuations.

Under the common assumption of a light neutrino exchange mechanism, the $^{130}$Te half-life limit converts to a limit on the effective Majorana mass of $m_{\beta\beta} < 90$–305 meV, with the spread induced by different nuclear matrix element calculations[34–40]. This limit on $m_{\beta\beta}$ is among the strongest in the field, proving the competitiveness of the cryogenic calorimeter technique used in CUORE. CUORE will continue to take data until it reaches its designed $^{130}$Te exposure of 1,000 kg yr.

## Impact

We have demonstrated that the cryogenic calorimeter technique is scalable to tonne-scale detector masses and multi-year measurement campaigns, while maintaining low radioactive backgrounds. Next-generation calorimetric 0$\nu\beta\beta$ decay searches exploiting these developments are planned. Among these, CUPID (CUORE Upgrade with Particle IDentification)[41] will use the same cryogenic infrastructure as CUORE, replacing the TeO$_2$ crystals with scintillating Li$_2^{100}$MoO$_4$ crystals and exploiting the scintillation light for greater than 100-fold active suppression of the $\alpha$ background[42,43]. In parallel, the AMoRE collaboration aims to build a large-mass calorimetric 0$\nu\beta\beta$ decay experiment in Korea[44]. In general, the possibility to cool large detector payloads paired with the low energy thresholds achievable by cryogenic calorimeters will benefit next-generation projects at the frontier of particle physics, for example dark matter searches such as Super-CDMS[45] and CRESST[46], and low-energy observatories exploiting CE$\nu$NS for solar and supernova neutrino studies[47] and neutrino flux monitoring of nuclear reactors[48].

A serendipitous effect is that the cryogenic innovations pioneered by CUORE for 0$\nu\beta\beta$ decay appear to be a solution-in-waiting for the challenges faced by the relatively young, but rapidly growing, field of quantum information technology. The need to cool increasingly large arrays of qubits to less than approximately 100 mK means there is now a commercial market for large, high-cooling-power dilution refrigerators, with some featuring technological solutions derived from CUORE. Moreover, the recent realization that ionizing radiation from natural radioactivity will be a limiting factor for the coherence time of quantum processors with an increasing number of qubits[49] suggests that the one-time niche, low-radioactivity ultralow-temperature cryogenics pioneered for 0$\nu\beta\beta$ decay may become mainstream in large-scale quantum computing[50].

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

**The CUORE Collaboration**

D. Q. Adams[1], C. Alduino[1], K. Alfonso[2], F. T. Avignone III[1], O. Azzolini[3], G. Bari[4], F. Bellini[5,6], G. Benato[7], M. Beretta[8], M. Biassoni[9], A. Branca[9,10], C. Brofferio[9,10], C. Bucci[7✉], J. Camilleri[11], A. Caminata[12], A. Campani[12,13], L. Canonica[7,14], X. G. Cao[15], S. Capelli[9,10], L. Cappelli[7], L. Cardani[6], P. Carniti[9,10], N. Casali[6], E. Celi[7,17], D. Chiesa[9,10], M. Clemenza[9,10], S. Copello[12,13], O. Cremonesi[9], R. J. Creswick[1], A. D'Addabbo[7,17], I. Dafinei[6], S. Dell'Oro[9,10], S. Di Domizio[12,13], V. Dompè[7,17], D. Q. Fang[15], G. Fantini[5,6], M. Faverzani[9,10], E. Ferri[9,10], F. Ferroni[6,17], E. Fiorini[9,10], M. A. Franceschi[18], S. J. Freedman[8,16,31], S. H. Fu[15], B. K. Fujikawa[16], A. Giachero[9,10], L. Gironi[9,10], A. Giuliani[19], P. Gorla[7], C. Gotti[9], T. D. Gutierrez[20], K. Han[21], E. V. Hansen[8], K. M. Heeger[22], R. G. Huang[8], H. Z. Huang[22], J. Johnston[14], G. Keppel[3], Yu. G. Kolomensky[8,16], C. Ligi[18], R. Liu[22], L. Ma[2], Y. G. Ma[15], L. Marini[7,8,16,17], R. H. Maruyama[22], D. Mayer[14], Y. Mei[16], N. Moggi[4,23], S. Morganti[6], T. Napolitano[18], M. Nastasi[9,10], J. Nikkel[22], C. Nones[24], E. B. Norman[25,26], A. Nucciotti[9,10], I. Nutini[9,10], T. O'Donnell[11], J. L. Ouellet[14], S. Pagan[22], C. E. Pagliarone[7,27], L. Pagnanini[7,17], M. Pallavicini[12,13], L. Pattavina[7], M. Pavan[9,10], G. Pessina[9], V. Pettinacci[6], C. Pira[3], S. Pirro[7], S. Pozzi[9,10], E. Previtali[9,10], A. Puiu[7,17], C. Rosenfeld[1], C. Rusconi[1,7], M. Sakai[8], S. Sangiorgio[25], B. Schmidt[16], N. D. Scielzo[25], V. Sharma[11], V. Singh[8], M. Sisti[9], D. Speller[28], P. T. Surukuchi[22], L. Taffarello[29], F. Terranova[9,10], C. Tomei[6], K. J. Vetter[8,16], M. Vignati[5,6], S. L. Wagaarachchi[8,16], B. S. Wang[25,26], B. Welliver[16], J. Wilson[1], K. Wilson[1], L. A. Winslow[14], S. Zimmermann[30] & S. Zucchelli[4,23]

[1]Department of Physics and Astronomy, University of South Carolina, Columbia, SC, USA. [2]Department of Physics and Astronomy, University of California, Los Angeles, Los Angeles, CA, USA. [3]INFN – Laboratori Nazionali di Legnaro, Legnaro, Italy. [4]INFN – Sezione di Bologna, Bologna, Italy. [5]Dipartimento di Fisica, Sapienza Università di Roma, Rome, Italy. [6]INFN – Sezione di Roma, Rome, Italy. [7]INFN – Laboratori Nazionali del Gran Sasso, Assergi, Italy. [8]Department of Physics, University of California, Berkeley, Berkeley, CA, USA. [9]INFN – Sezione di Milano Bicocca, Milan, Italy. [10]Dipartimento di Fisica, Università di Milano-Bicocca, Milan, Italy. [11]Center for Neutrino Physics, Virginia Polytechnic Institute and State University, Blacksburg, VA, USA. [12]INFN – Sezione di Genova, Genova, Italy. [13]Dipartimento di Fisica, Università di Genova, Genova, Italy. [14]Massachusetts Institute of Technology, Cambridge, MA, USA. [15]Key Laboratory of Nuclear Physics and Ion-beam Application (MOE), Institute of Modern Physics, Fudan University, Shanghai, China. [16]Nuclear Science Division, Lawrence Berkeley National Laboratory, Berkeley, CA, USA. [17]Gran Sasso Science Institute, L'Aquila, Italy. [18]INFN – Laboratori Nazionali di Frascati, Frascati, Italy. [19]IJCLab, Université Paris-Saclay, CNRS/IN2P3, Orsay, France. [20]Physics Department, California Polytechnic State University, San Luis Obispo, CA, USA. [21]Shanghai Laboratory for Particle Physics and Cosmology, INPAC, School of Physics and Astronomy, Shanghai Jiao Tong University, Shanghai, China. [22]Wright Laboratory, Department of Physics, Yale University, New Haven, CT, USA. [23]Dipartimento di Fisica e Astronomia, Alma Mater Studiorum – Università di Bologna, Bologna, Italy. [24]IRFU, CEA, Université Paris-Saclay, Gif-sur-Yvette, France. [25]Lawrence Livermore National Laboratory, Livermore, CA, USA. [26]Department of Nuclear Engineering, University of California, Berkeley, CA, USA. [27]Dipartimento di Ingegneria Civile e Meccanica, Università degli Studi di Cassino e del Lazio Meridionale, Cassino, Italy. [28]Department of Physics and Astronomy, The Johns Hopkins University, Baltimore, MD, USA. [29]INFN – Sezione di Padova, Padova, Italy. [30]Engineering Division, Lawrence Berkeley National Laboratory, Berkeley, CA, USA. [31]Deceased: S. J. Freedman. ✉e-mail: cuore-spokesperson@lngs.infn.it

# Methods

## Optimum trigger and analysis threshold

The continuous data stream of CUORE is triggered with the optimum trigger, an algorithm based on the optimum filter[51] characterized by a lower threshold than a more standard derivative trigger[32]. A lower threshold enables us not only to reconstruct the low-energy part of the spectrum, but also yields a higher efficiency in reconstructing the events in coincidence between different calorimeters, and consequently a more precise understanding of the corresponding background components[52,53].

The optimum trigger transfer function of every event is matched to the ideal signal shape, obtained as the average of good-quality pulses, so that frequency components with low signal-to-noise ratio are suppressed. A trigger is fired if the filtered signal amplitude exceeds a fixed multiple of the noise root mean square (RMS), defined separately for each calorimeter and dataset. We evaluate the energy threshold by injecting fake pulses of varying amplitude, calculated by inverting the calibration function, into the data stream. We reconstruct the stabilized amplitude of the fake pulses, fit the ratio of correctly triggered events to generated events with an error function, and use the 90% quantile as a figure of merit for the optimum trigger threshold. This approach enables monitoring of the threshold during data collection, and is based on the assumption that the signal shape is not energy dependent, that is, that the average pulse obtained from high-energy $\gamma$ events is also a good template for events of a few keV. The distribution of energy threshold at 90% trigger efficiency is shown in Extended Data Fig. 4.

For this work we set a common analysis threshold of 40 keV, which results in >90% trigger efficiency for the majority (97%) of the calorimeters, while at the same time allowing the removal of multi-Compton events from the region of interest through the anti-coincidence cut.

## Efficiencies

The total efficiency is the product of the reconstruction, anti-coincidence, pulse shape discrimination (PSD) and containment efficiencies.

The reconstruction efficiency is the probability that a signal event is triggered, has the energy properly reconstructed, and is not rejected by the basic quality cuts requiring a stable pre-trigger voltage and only a single pulse in the signal window. It is evaluated for each calorimeter using externally flagged heater events[54], which are a good approximation of signal-like events.

The anti-coincidence efficiency is the probability that a true single-crystal event correctly passes our anti-coincidence cut, instead of being wrongly vetoed owing to an accidental coincidence with an unrelated event. It is extracted as the acceptance of fully absorbed $\gamma$ events at 1,460 keV from the electron capture decays of $^{40}$K, which provide a reference sample of single-crystal events.

The PSD efficiency is obtained as the average acceptance of events in the $^{60}$Co, $^{40}$K and $^{208}$Tl $\gamma$ peaks that already passed the base and anti-coincidence cuts. In principle, the PSD efficiency could be different for each calorimeter, but given the limited statistics in physics data we evaluate it over all channels and over the full dataset. To account for possible variation between individual calorimeters, we compare the PSD efficiency obtained by directly summing their individual spectra with that extracted from an exposure-weighted sum of the calorimeters' spectra. We find an average ±0.3% discrepancy between the two values and include it as a global systematic uncertainty in the $0\nu\beta\beta$ fit. This takes a Gaussian prior instead of the uniform prior used in our previous result[32], which had its uncertainty come from a discrepancy between two approaches that has since been resolved.

Finally, the containment efficiency is evaluated through Geant4-based Monte Carlo simulations[55] and accounts for the energy loss due to geometrical effects as well as bremsstrahlung.

## Principal component analysis for PSD

In this analysis we use a new algorithm based on principal component analysis (PCA) for pulse shape discrimination. The method has been developed and documented for CUPID-Mo[56], and has been adapted for use in CUORE[57]. This technique replaces the algorithm used in previous CUORE results, which was based on six pulse shape variables[30]. The PCA decomposition of signal-like events pulled from $\gamma$ calibration peaks yields a leading component similar to an average pulse, which on its own captures >90% of the variance between pulses. We choose to treat the average pulse of each calorimeter in a dataset as if it were the leading PCA component, normalizing it like a PCA eigenvector. We can then project any event from the same channel onto this vector and attempt to reconstruct the 10-s waveform using only this leading component. For any waveform $\mathbf{x}$ and leading PCA component $\mathbf{w}$ with length $n$, we define the reconstruction error as:

$$\mathrm{RE} = \sqrt{\sum_{i=1}^{n} (\mathbf{x}_i - (\mathbf{x} \cdot \mathbf{w})\mathbf{w}_i)^2}. \tag{1}$$

This reconstruction error metric measures how well an event waveform can be reconstructed using only the average pulse treated as a leading PCA component. Events that deviate from the typical expected shape of a signal waveform are poorly reconstructed and have a high reconstruction error. We normalize the reconstruction errors as a second-order polynomial function of energy on a calorimeter-dataset basis (see Extended Data Fig. 5), and cut on the normalized values by optimizing a figure of merit for signal efficiency over expected background in the $Q_{\beta\beta}$ region of interest. Using this PCA-based method, we obtain an overall efficiency of (96.4 ± 0.2)% compared to the (94.0 ± 0.2)% from the pulse shape analysis used in our previous results, as well as a 50% reduction in the PSD systematic uncertainty from 0.6% to 0.3%.

## Statistical analysis

The high-level statistical $0\nu\beta\beta$ decay analysis consists of an unbinned Bayesian fit on the combined data developed with the BAT software package[58]. The model parameters are the $0\nu\beta\beta$ decay rate ($\Gamma_{0\nu}$), a linearly sloped background, and the $^{60}$Co sum peak amplitude. $\Gamma_{0\nu}$ and the $^{60}$Co rate are common to all datasets, with the $^{60}$Co rate scaled by a preset dataset-dependent factor to account for its expected decay over time. The base background rate, expressed in terms of counts keV$^{-1}$ kg$^{-1}$ yr$^{-1}$, is dataset-dependent, whereas the linear slope to the background is shared among all datasets, because any structure to the shape of the background should not vary between datasets. $\Gamma_{0\nu}$, the $^{60}$Co rate, and the background rate parameters have uniform priors that are constrained to non-negative values, whereas the linear slope to the background has a uniform prior that allows both positive and negative values.

In addition to these statistical parameters, we consider the systematic effects induced by the uncertainty on the energy bias and energy resolution[59,60], the value of $Q_{\beta\beta}$, the natural isotopic abundance of $^{130}$Te, and the reconstruction, anti-coincidence, PSD and containment efficiencies. We evaluate their separate effects on the $0\nu\beta\beta$ rate by adding nuisance parameters to the fit one at a time and studying both the effect on the posterior global mode $\hat{\Gamma}_{0\nu}$ and the marginalized 90% CI limit on $\Gamma_{0\nu}$.

A list of the systematics and priors is reported in Extended Data Table 1. The efficiencies and the isotopic abundance are multiplicative terms on our expected signal, so the effect of each is reported as a relative effect on $\Gamma_{0\nu}$. By contrast, the uncertainties on $Q_{\beta\beta}$, the energy bias, and the resolution scaling have a non-trivial relation to the signal rate; therefore, we report the absolute effect of each on $\Gamma_{0\nu}$. The dominant effect is due to the uncertainty on the energy bias and resolution scaling in physics data. We account for possible correlations between the nuisance parameters by including all of them in the fit simultaneously.

We chose a uniform prior on our physical observable of interest $\Gamma_{0\nu}$, which means we treat any number of signal events as equally likely.

Other possible uninformative choices could be considered appropriate, as well. Because the result of any Bayesian analysis depends to some extent on the choice of the priors, we checked how other reasonable priors affect our result[57]. We considered: a uniform prior on $\sqrt{\Gamma_{0\nu}}$, equivalent to a uniform prior on $m_{\beta\beta}$ and also equivalent to using the Jeffreys prior; a scale-invariant uniform prior on $\log\Gamma_{0\nu}$; and a uniform prior on $1/\Gamma_{0\nu}$, equivalent to a uniform prior on $T_{1/2}^{0\nu}$.

These priors are all undefined at $\Gamma_{0\nu} = 0$, so we impose a lower cut-off of $\Gamma_{0\nu} > 10^{-27}$ yr$^{-1}$, which with the given exposure corresponds to approximately one signal event. The case with a uniform prior on $\sqrt{\Gamma_{0\nu}}$ gives the smallest effect, and strengthens the limit by 25%, whereas the flat prior on $1/\Gamma_{0\nu}$ provides the largest effect, increasing the limit on $T_{1/2}^{0\nu}$ by a factor of 4. In fact, all these priors weigh the small values of $\Gamma_{0\nu}$ more. Therefore, our choice of a flat prior on $\Gamma_{0\nu}$ provides the most conservative result.

We compute the $0\nu\beta\beta$ exclusion sensitivity by generating a set of $10^4$ toy experiments with the background-model, that is, including only the $^{60}$Co and linear background components. The toys are split into 15 datasets with exposure and background rates obtained from the background-only fits to our actual data. We fit each toy with the signal-plus-background model, and extract the distribution of 90% CI limits, shown in Extended Data Fig. 4.

We perform the frequentist analysis using the Rolke method[61], obtaining a lower limit on the process half-life of $T_{1/2}^{0\nu} > 2.6 \times 10^{25}$ yr (90% CI). The profile likelihood function $\mathcal{L}$ for $\Gamma_{0\nu}$ is retrieved from the full Markov chain produced by the Bayesian analysis tool. The non-uniform priors on the systematic effects in the Bayesian fit are thus incorporated into the frequentist result as well. We extract a 90% confidence interval on $\Gamma_{0\nu}$ by treating $-2\log\mathcal{L}$ as an approximate $\chi^2$ distribution with one degree of freedom. The lower limit on $T_{1/2}^{0\nu}$ comes from the corresponding upper edge of the confidence interval on $\Gamma_{0\nu}$. Applying the same method to the toy experiments, we find a median exclusion sensitivity of $T_{1/2}^{0\nu} > 2.9 \times 10^{25}$ yr.

## Data availability

The data generated during this analysis and shown in paper figures are available in ASCII format (CSV) as Source Data in the repository https://cuore.lngs.infn.it/en/publications/collaborationpapers. Additional information is available upon request by contacting the CUORE Collaboration.

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

**Acknowledgements** We thank the directors and staff of the Laboratori Nazionali del Gran Sasso and the technical staff of our laboratories. This work was supported by the Istituto Nazionale di Fisica Nucleare (INFN); the National Science Foundation under grant nos. NSF-PHY-0605119, NSF-PHY-0500337, NSF-PHY-0855314, NSF-PHY-0902171, NSF-PHY-0969852, NSF-PHY-1614611, NSF-PHY-1307204, NSF-PHY-1314881, NSF-PHY-1401832 and NSF-PHY-1913374; and Yale University. This material is also based upon work supported by the US Department of Energy (DOE) Office of Science under contract nos. DE-AC02-05CH11231 and DE-AC52-07NA27344; by the DOE Office of Science, Office of Nuclear Physics under contract nos. DE-FG02-08ER41551, DE-FG03-00ER41138, DE-SC0012654, DE-SC0020423, DE-SC0019316; and by the EU Horizon 2020 research and innovation programme under Marie Skłodowska-Curie Grant agreement no. 754496. This research used resources of the National Energy Research Scientific Computing Center (NERSC). This work makes use of both the DIANA data analysis and APOLLO data-acquisition software packages, which were developed by the CUORICINO, CUORE, LUCIFER and CUPID-0 collaborations.

**Author contributions** All listed authors have contributed to the present publication. The different contributions span from the design and construction of the detector and of the cryogenic system to the acquisition and analysis of data. The manuscript underwent an internal review process extended to the whole collaboration, and all authors approved its final version; the authors' names are listed alphabetically.

**Competing interests** The authors declare no competing interests.

**Additional information**
**Correspondence and requests for materials** should be addressed to C. Bucci.

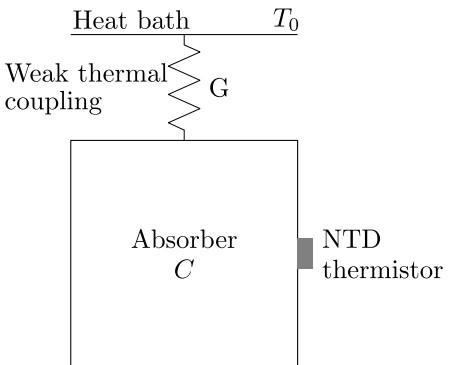

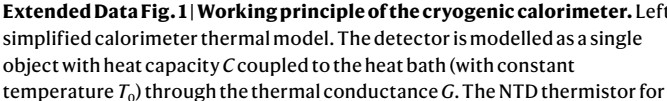

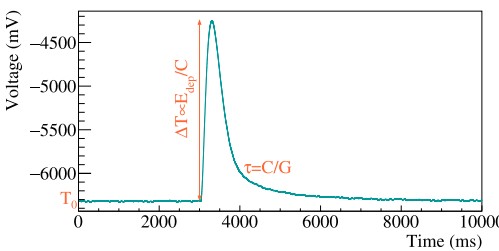

**Extended Data Fig. 1 | Working principle of the cryogenic calorimeter.** Left, simplified calorimeter thermal model. The detector is modelled as a single object with heat capacity $C$ coupled to the heat bath (with constant temperature $T_0$) through the thermal conductance $G$. The NTD thermistor for signal readout is glued to the absorber. Right, example of a CUORE pulse from the 2,615-keV calibration line: $T_0$ corresponds to the baseline height, the pulse amplitude is proportional to the deposited energy, and the decay time depends on the value of $C/G$.

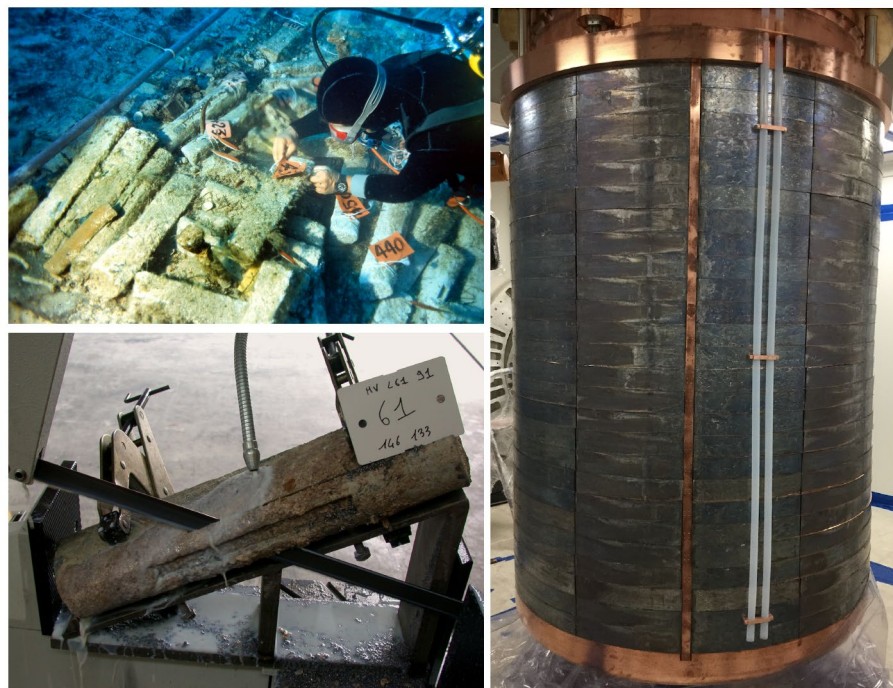

**Extended Data Fig. 2 | Roman lead.** Top left, the recovery of the lead bricks from the Sardinian sea. Bottom left, the ingot inscriptions were cut and preserved, and the ingot bodies were used for the CUORE internal lead shield. Right, lateral view of the internal lead shield[20].

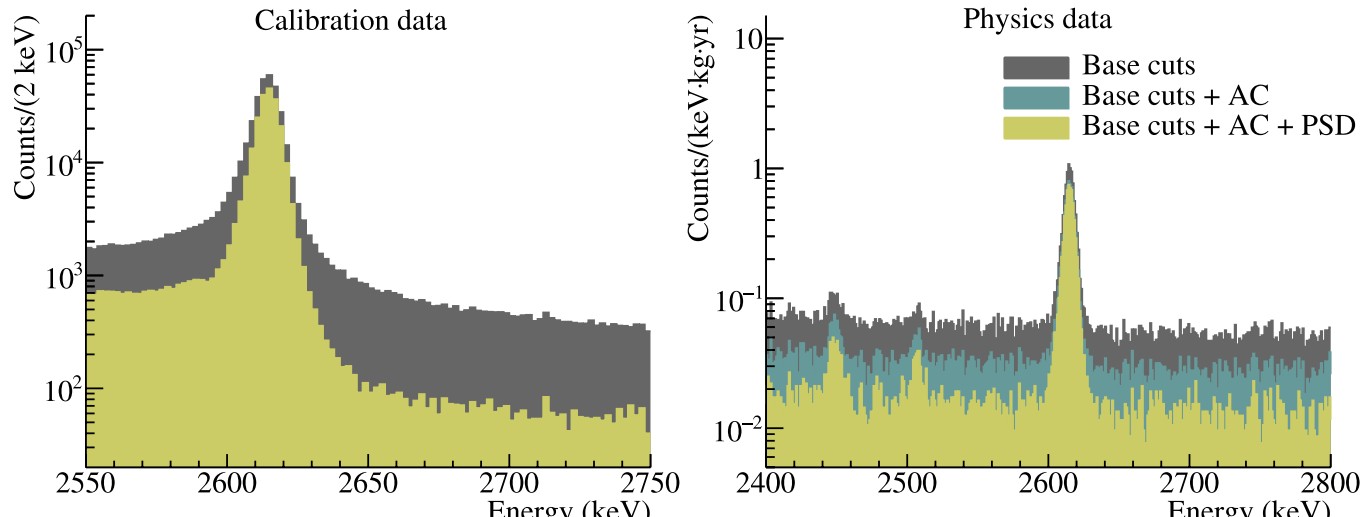

**Extended Data Fig. 3 | Pulse shape discrimination.** Effect of the PSD cut on calibration data around the 2,615-keV line (left) and on physics data near $Q_{\beta\beta}$ (right). In calibration data, the anti-coincidence is not applied in order to maximize the statistics on the γ peaks, and the PSD mostly removes pileup events (events with more than one energy deposit in the time window).

In physics data, the PSD mostly eliminates random noise events, which can correspond to either physical events with excessive noise or to noise-induced events with non-physical pulse shapes. Such events appear randomly across the energy spectrum, so the cut mostly acts on the continuum.

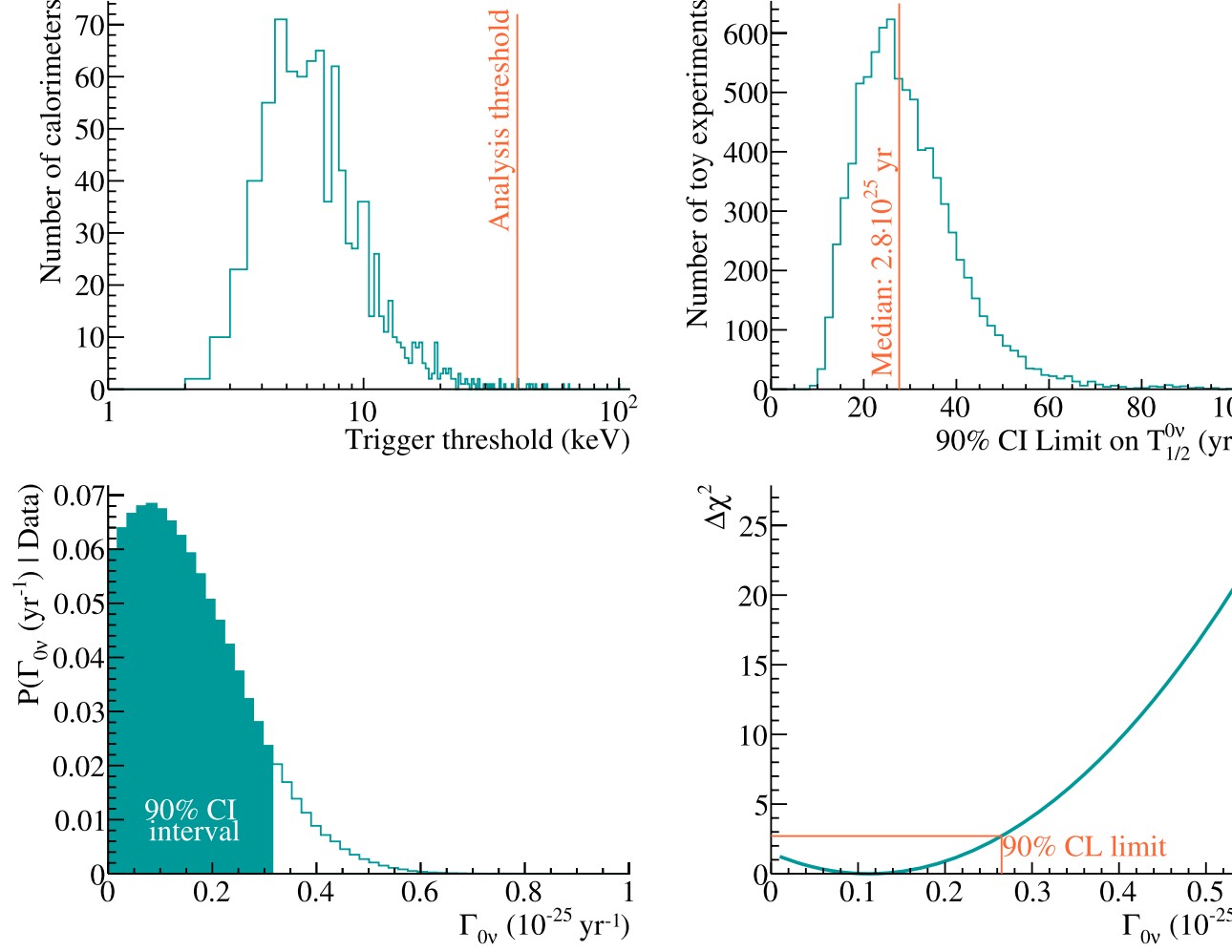

**Extended Data Fig. 4 | Optimum trigger and statistical analysis.** Top left, distribution of energy thresholds at 90% trigger efficiency for the optimum trigger algorithm in a single dataset. The 40-keV analysis threshold is indicated by the vertical line. Top right, 90% CI exclusion limits on $T_{1/2}^{0\nu}$ from an ensemble of $10^4$ toy experiments generated with the background-only model, with background rates obtained from the background-only fit to the data. The median exclusion sensitivity is indicated by the orange line. Bottom left, posterior probability distribution for $\Gamma_{0\nu}$ obtained from the Bayesian fit, with the 90% CI highlighted. Bottom right, $\Delta\chi^2$ values obtained from the profile likelihood of $\Gamma_{0\nu}$, with $\Delta\chi^2 = 0$ being the most-favoured value. The frequentist limit at 90% confidence level (CL) is indicated.

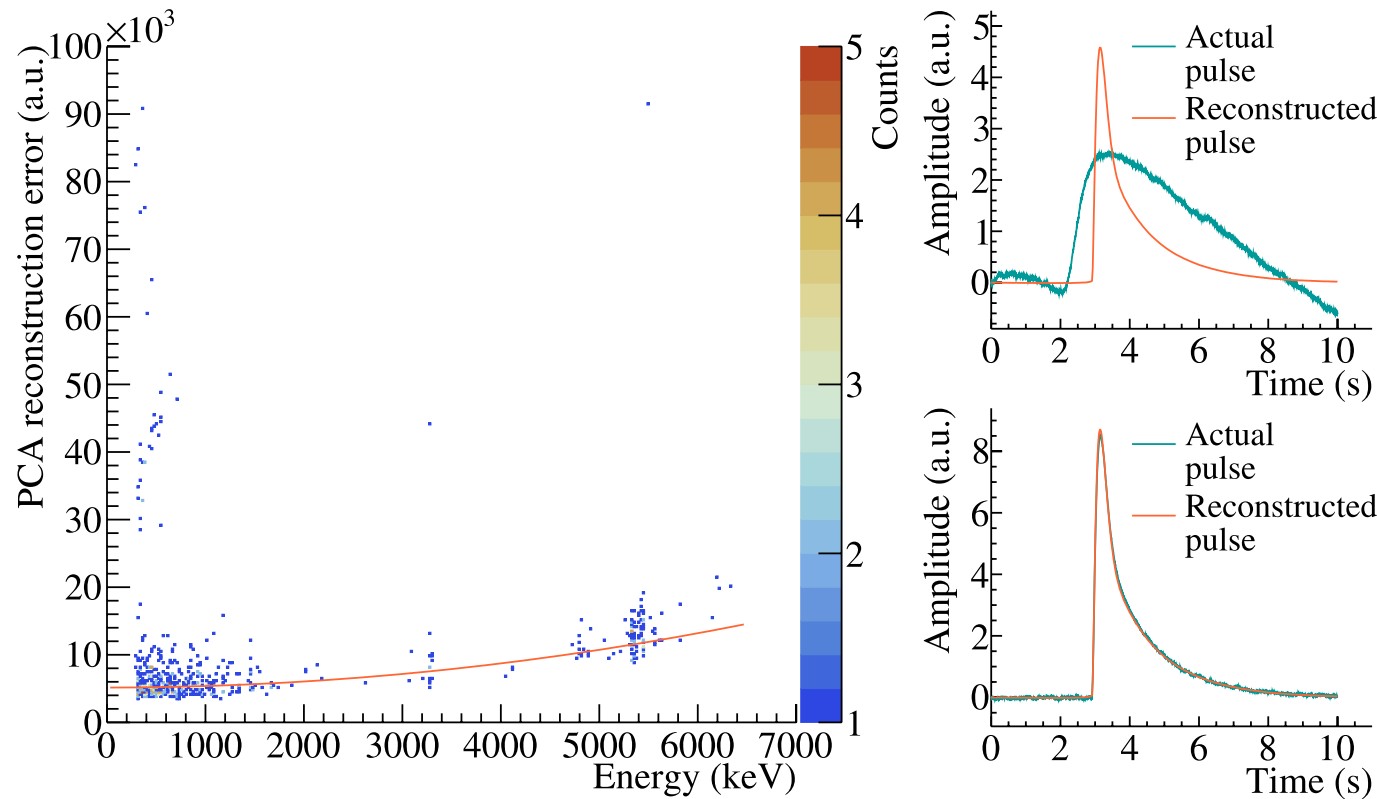

**Extended Data Fig. 5 | PCA performance.** Left, example of a normalization fit of the PCA reconstruction error versus energy for a single calorimeter and dataset. The distribution contains only events that passed the other base cuts. The second-order polynomial fit is shown in orange. Right, two example pulses from this calorimeter. The actual pulse is drawn in teal, and the corresponding reconstruction obtained by the PCA is drawn in orange. The top pulse deviates from the expected shape of a good pulse and is rejected, whereas the bottom one conforms to the expected response and is accepted.

**Extended Data Table 1 | Systematics affecting the 0νββ decay analysis**

| Fit parameter systematics | | | |
|---|---|---|---|
| Systematic | Prior | Effect on the Marginalized $\Gamma_{0\nu}$ Limit | Effect on $\hat{\Gamma}_{0\nu}$ |
| Total analysis efficiency | Gaussian | 0.2% | < 0.1% |
| PSD efficiency | Gaussian | 0.3% | < 0.1% |
| Containment efficiency | Gaussian | 0.2% | < 0.1% |
| Isotopic abundance | Gaussian | 0.2% | < 0.1% |
| $Q_{\beta\beta}$ | Gaussian | $< 0.1 \cdot 10^{-27}$ yr$^{-1}$ | $< 0.1 \cdot 10^{-27}$ yr$^{-1}$ |
| Energy bias and Resolution scaling | Multivariate | $0.2 \cdot 10^{-27}$ yr$^{-1}$ | $0.1 \cdot 10^{-27}$ yr$^{-1}$ |

The total analysis efficiency is the product of all the efficiencies listed in Table 1 except containment. The PSD efficiency refers to its additional systematic uncertainty described in the text. The first four systematics are multiplicative effects and the impact of each is presented as a percentage. The last two systematics have a non-trivial effect on $\Gamma_{0\nu}$, hence we report the absolute effect. We report the variation induced on the marginalized 90% CI limit (third column) and the posterior global mode $\hat{\Gamma}_{0\nu}$ (final column).