## [Peer Review File · Nature]

Manuscript Title: Search for Majorana neutrinos exploiting milli-Kelvin cryogenics with CUORE

Reviewer Comments & Author Rebuttals

Reviewer Reports on the Initial Version:

Referee #1:

The paper titled "High sensitivity neutrinoless double-beta decay search with one tonne-year of CUORE data" summarizes the latest data release from CUORE and presents a new half-life limit with a three-fold exposure increase compared to previous results. The manuscript provides original results that is considered highly significant in the field of rare event searches in particle physics. However in my opinion, it lacks the relevance for other fields in physics or other disciplines.

The description of the data and methodology is understandable with minor clarifications to the calibrations detailed in my questions below. The statistical analysis with both Bayesian and frequentist methods is commendable, for the more interested readers some additional details could be included in the method section including some clarifications. The conclusions of the paper and the results are valid and reliable. The further developments in the search for neutrinoless double-beta decay with a more sensitive cryogenic experiment are well motivated.

Unfortunately, the text has quite some english mistakes, especially mixing plural and singular that makes reading sometimes difficult. Below I list my questions and comments to the content including the Methods and then separately some suggestions to the language and style of the presentation.

Comments and questions

- l179 where are the external sources positioned?
- l196-213 the two paragraphs are a bit confusing: how large are these stabilization corrections? how long is a calibration that time instabilities are important? how many heater pulses are taken during one calibration? how many detectors have an unstable heater? how is the stabilisation done for physics data for detectors with unstable heater?
- table 1 is missing the Te-130 exposure
- l328 the signal shape is described to come from calibration with 3 gaussians but it would be good to have in the main text the background shape that goes into the fit
- how does the signal line shape influence the fit results and the limit?
- is the linear background shape supported by the background model?
- l739 stabilization is not clear, and why is it necessary here?
- l792 can you comment on the dependence of the fit result on the choice of priors?
- l838 do you include the Bayesian priors into the frequentist fit? the description of the frequentist analysis should have more details or at least a reference to the Rolke method

Language and style

- general: plural and singular are incorrect in some places
- abstract: inconsistent universe vs Universe
- l19 for non-physicists neutrino families and flavors might not be clearly the same thing
- it is a matter of taste but somehow the historical introduction reads like a bunch of statements without much connection
- l37 neutrino capable sounds like it can decide about its nature
- l44 very strong statement "only realistic", there are efforts in HEP just more model dependent
- fig 1 colors not visible in black and white

- 1186 how many datasets are there in the end? found only in table 1
- 1284 response function is a bit unusual formulation for the gamma line shape
- 1734 add +for+ any physics run  run not defined, is it dataset?
- 1738 this description of the trigger efficiency procedure should be before the discussion on advantages and disadvantages because now it is hard to read
- 1776 the sentence has no end

In summary, I find that this paper after a major editorial revision should be published, but it would better fit in a physics journal, than Nature because of lack of interest from other fields.

Referee #2:

A. Summary of the key results

This paper is an update from CUORE's 2020 PRL with ~triple the exposure, reaching 1 t-y of exposure of TeO₂. Their neutrinoless double-beta decay limit comes out similar to the previous result but the sensitivity is a ~60% improvement.

B. Originality and significance: if not novel, please include reference

This is the article's weakpoint in my opinion. The CUORE technique is definitely original / novel compared to what came before it, and their limit is the best for ¹³⁰Te. However, this is not CUORE's first publication, and its originality and significance is weak especially compared to their article in Physics Review Letters just last year. Nature is usually in the business of publishing ground-breaking results, however to me this seems more like a minor data update. The scientific effort involved in this specific paper appears to have essentially involved simply continuing to take data, and turning the analysis crank with a few minor improvements.

Even the milestone the authors claim to have achieved -- a 1 ton-year exposure -- requires some hand waving. To get to 1 ton-year, CUORE counts the full mass of deployed TeO₂ crystals. This includes both the double-beta decay isotope ¹³⁰Te and its "carrier materials" including oxygen. By this accounting, the CUORE exposure is far exceeded by other neutrinoless double-beta decay experiments, notably KamLAND-Zen with a 19 t-y Xe-LS exposure published in 2016. Of course what matters for the $0\nu\beta\beta$ result is just the mass in the active isotope, ¹³⁰Te. This exposure is only 1/3 of a ton year, which is also exceeded for example again by the KamLAND-Zen 2016 result, which achieved a 0.5 t-y isotopic exposure.

And CUORE is hardly the first ton-scale "cryogenic particle physics" experiment. It's not even the first one in LNGS, with the XENON1T WIMP Dark Matter detector running right down the hall from them. XENON1T is not nearly as cold as CUORE, still just last year they published a result that by CUORE accounting was a 2-ton-year exposure. It only falls below 1 ton year when you account for fiducialization, the role of which in this paper is played by the 88% containment efficiency, which would also bring CUORE below the 1 t-y threshold highlighted in the article's title.

For sure there is still a significant technical achievement here. I think the authors are trying to express that CUORE is perhaps the most technically challenging experiment of its kind performed to date (large-mass calorimetric experiment operated at sub-liquid-He temperatures), and 1 ton-year of exposure is indeed an achievement, even if it's an arbitrary goal post. However there are of course many most-challenging-of-its-kind-ever experiments, modern examples that comes to mind are ATLAS/CMS, or within neutrino physics, for example KATRIN. In the end, one can always put enough qualifiers in a sentence to claim to have been the first to do something. The question is whether it's important. And even here the article spends almost no time arguing how CUORE's technical achievement is important. It contains essentially no broader or historical context for how or why 1 t-y of TeO₂ exposure is so much more important than their own 300 t-y last year, or 19 t-y or 2 t-y in other experiments. There is only a sentence or two about relevance to future $0\nu\beta\beta$

experiments and an extremely vague statement name-dropping quantum computing with no explanation or further discussion.

I'm a big fan of $0\nu\beta\beta$ physics and like to see it getting headlines in high-profile journals. And I do congratulate the CUORE team on their excellent result. However I have to admit that ultimately I'm surprised that this specific result is being considered for publication in Nature.

C. Data & methodology: validity of approach, quality of data, quality of presentation

- Line 212 and 303: Why does the energy scale require a 2nd order polynomial fit? What is the source of the nonlinearity?

- It was unclear to me how precisely the salting method works. Is it applied at the level of raw data? That seems difficult since the raw data is usually integer in nature, so scaling by a factor of 2527 / 2615 would make things obviously non-integer and detectable by analysts. Conversely it could be done at the energy reconstruction stage, but then it seems one could simply re-analyze the raw data and conclude whether an event was "salt" or not. Some better explanation would help.

- The PSD method described in lines 236-240 is claimed to remove pileup events and non-physical pulses, yet the Figure 2 lower-right panel shows it to also affect the calibration peak width. I suspect the PSD is also being used to remove "noisy waveforms" as is done in the 2020 PRL. If this is the case, the description should be updated to reflect this.

- On the same topic -- the PSD does not seem to also sharpen the peaks in the physics data shown in the lower-left plot of Fig 2. Is this a problem for the resolution function model? Some comment should be added on this. See also my lengthy comments on the resolution model below.

- The selection criteria paragraph starting on line 223 does not describe the "basic quality cuts" mentioned in line 257. What are these? Their description should be added somewhere.

- The last sentence of p3 states that the reconstruction efficiency is computed by-calorimeter and then averaged over the entire dataset. Why is this necessary? Is it highly variable? If so, why?

- Fig 2, lower-left panel: the PSD removes much more continuum (factor of ~ 2) than peak. Why is this the case? It's hard to believe that it's due to pileup, the effect of which is already only a factor of ~ 2 reduction in the vastly higher-rate calibration data shown in the lower-right plot. Some comment on this should be added.

- In the paragraph starting on line 271, it's described that physics data is used to extract the PSD efficiency, and statistics are too low to allow for a by-calorimeter measurement. Why can't the PSD efficiency be extracted directly from high-stats calibration data?

- I was particularly concerned about the detector response handling. Does the same triple-gaussian parameterization used for the 2615 calibration peak work for all peaks across all energies? The references only characterize FWHM(E) but I could not tell whether e.g. the parameters of each gaussian are varying in a sensible way across E. More importantly, it seems perfectly likely that the non-gaussian behavior is associated with the position distribution of energy depositions in the crystals. $0\nu\beta\beta$ is very different in this respect from 2615 keV gamma rays, so I worry that the uncertainty in the $0\nu\beta\beta$ peak shape is underestimated, which would impact the final extracted limits.

The resolution function parameterization is not just important for the $0\nu\beta\beta$ peak, but also for the ^{60}Co sum peak where there are 2 gammas and probably a much broader spatial distribution. This one is so important because its proximity to $Q_{\beta\beta}$ (anti)correlates the peak width with the extracted $0\nu\beta\beta$ rate. So the resolution function comes into the final result effectively twice, for two peaks with fundamentally different energy deposition topologies from each other and from the peak used to characterize the empirical resolution function. It's thus critical to have some demonstration on whether or not the empirical model works for all event topologies.

In addition to checking ^{60}Co (in calibration and physics data), I recommend also doing the same test on other peaks from interactions with fundamentally different topologies. For example, double-escape peak events in the calibration data have a much more localized energy deposition, although positron annihilation-in-flight leads to some extra width that can be estimated with MC. The single escape peak provides another good test case. Low-energy peaks are also more single-site and more localized near detector surfaces and would provide another important test case.

- It was also concerning to me that in lines 305-306 the manuscript described the variation of FWHM with energy to be linear, while the PhD theses referenced in the article both appear to show non-linear trends. Some comment or demonstration of linearity should be added.

D. Appropriate use of statistics and treatment of uncertainties

The Bayesian limit uses a flat prior on the $0\nu\beta\beta$ rate with a physical boundary imposed to exclude negative rates. Such limits are notoriously prior-dependent. The sensitivity to choice of prior for the $0\nu\beta\beta$ rate should be evaluated and reported (in the Methods section).

The frequentist limit description is missing a reporting of the median sensitivity, this should be added to the last paragraph in the Methods section. It would also be good to report the frequentist χ^2 in Fig. 3 (for example with a separate right-axis scale).

E. Conclusions: robustness, validity, reliability

The Discussion section points out that the limit is weaker than in the 2020 PRL. To aid in this explanation, the authors might also point out that the previous limit was 2x better than its sensitivity -- well within expected fluctuations, but made it more likely that the next result (this limit) would be weaker.

F. Suggested improvements: experiments, data for possible revision

I recommend further study of the energy resolution function, see comments in (C)

G. References: appropriate credit to previous work?

- A reference is missing for the data salting method where one can find details on how it is done.
- A reference is missing for the Rolke method mentioned in line 837.

H. Clarity and context: lucidity of abstract/summary, appropriateness of abstract, introduction and conclusions

It's a bit strange that the abstract has 5-6 lines about leptogenesis, while the intro only mentions

it in the last sentence of the first paragraph after a long exposition on the history of neutrinos. In my opinion, the intro should be re-written to match the much more attention-grabbing abstract.

I also noticed a couple of instances of redundancy, which was surprising for such a short article:

- The containment efficiency is described twice, once near line 231 and again near line 253.
- The quadratic form of the energy scale was mentioned twice, once in 212 and again in 303, without reference to the earlier text. This might confuse the reader into thinking that they are two different things when as far as I can tell they are the same.

Referee #3:

The authors present a comprehensive analysis from the first tonne-year exposure of CUORE, searching for $0\nu\beta\beta$ decay of Te-130. While the half-life limit is in line with their previous result (albeit increased statistics), this is the world's most sensitive search for $0\nu\beta\beta$ decay with Te-130 and a technological milestone that paves the way for a much improved upgrade, called CUPID. I believe this technological achievement, along with an improved analysis method, merits publication in Nature.

Overall, the manuscript is well-written, well-organized, and a pleasure to read. However, I do have several comments that I hope the authors consider before the manuscript is accepted for publishing. They are as follows:

1) Lines 63-64: "...anti-coincidence efficiency is the probability that a signal event is not incorrectly vetoed due to an accidental coincidence between two independent events." Did you really mean to use a double negative here? Should it not be "correctly vetoed" instead of "not incorrectly vetoed"?

2) Extended Data Figure 8: The distribution of pseudo experiments shows the familiar shape we typically see in these kinds of results. However, when I follow the diminishing tail of the distribution towards longer half-life, I notice that the very last bin near 10^{26} years contains about 20 experiments. It's noticeably higher than all the preceding bins and almost makes the reader wonder if something strange is happening in the background-only fitting procedure. Of course, this small number of pseudo experiments will only have a very small effect on the median limit, but it would be good to confirm that this strange bin is not a symptom of some underlying problems with the fit.

3) Extended Data Figure 7: The table listing the various systematics has a column describing the different priors used. I notice that "Analysis efficiency II" referring to the PSD efficiency assumes a Gaussian prior. This is an important difference from the previous analysis reported in DOI: 10.1103/PhysRevLett.124.122501, where a uniform prior was assumed. I'd prefer to see a sentence or two in the body of the text explicitly describing this difference, since many readers will likely be comparing the present analysis with the previous one. Why wasn't this available as a Gaussian prior in the previous analysis?

4) Discussion Section: As the authors state in the Discussion section, the analysis presented in this manuscript is weaker than their previous result and is well within the expected range of outcomes due to statistical fluctuations. While statistical fluctuations are out of our control, the data analysis seems quite mature at this point and has the added improvement of PCA used to reject background with higher signal selection efficiency. It would be beneficial to highlight the maturity of the CUORE analysis at this stage in the experiment and provide readers with a statement concerning any further improvements in the analysis (absent the major hardware upgrade for CUPID). Is this CUORE analysis as good as it gets, and is the experiment now limited by backgrounds in the ROI?

Author Rebuttals to Initial Comments:

Questions and answers to referees

Referee #1 (Remarks to the Author):

The paper titled "High sensitivity neutrinoless double-beta decay search with one tonne-year of CUORE data" summarizes the latest data release from CUORE and presents a new half-life limit with a three-fold exposure increase compared to previous results. The manuscript provides original results that are considered highly significant in the field of rare event searches in particle physics. However in my opinion, it lacks the relevance for other fields in physics or other disciplines.

The description of the data and methodology is understandable with minor clarifications to the calibrations detailed in my questions below. The statistical analysis with both Bayesian and frequentist methods is commendable, for the more interested readers some additional details could be included in the method section including some clarifications. The conclusions of the paper and the results are valid and reliable. The further developments in the search for neutrinoless double-beta decay with a more sensitive cryogenic experiment are well motivated.

Unfortunately, the text has quite some English mistakes, especially mixing plural and singular that makes reading sometimes difficult. Below I list my questions and comments to the content including the Methods and then separately some suggestions to the language and style of the presentation.

- l179 where are the external sources positioned?

The sources are lowered down between the 300 K vessel and the external shielding.

- l196-213 The two paragraphs are a bit confusing: how large are these stabilization corrections?

How long is a calibration that time instabilities are important?

We extended the explanation to clarify the procedure, and further details on it are provided in the references indicated. Each source calibration period lasts a few days on average. We have two source calibration periods per dataset, one at the beginning and one at the end of the dataset. Datasets are about 1 month in duration for the ones acquired before 2019 and about 2 months in duration for the following since we achieved more stable operating conditions. We perform thermal gain stabilization continuously during the data taking by periodically injecting heater pulses. This allows us to correct for possible small drifts in the detector temperature over the duration of the months long dataset. Time instabilities during a single calibration period tend to be relatively insignificant, but since the calibration is applied to the whole dataset, the instabilities over the entire period are important and must

be taken into account.

How many heater pulses are taken during one calibration?

Heater pulses are fired simultaneously on all the channels of the same column. The period between consecutive pulses on a given bolometer is 580 s. Considering an average run of 24 hours, we have ~150 heater pulses per channel per run.

How many detectors have an unstable heater?

In CUORE we have 4 channels with non functioning thermistors and 29 channels with non functioning heaters. This means in principle we can employ the heater stabilization method for 97% of the active bolometers. Since heater-based stabilization is performed separately for each run, bolometers whose heater is unstable on a given run are discarded as well. This amounts to an average of ~12-40 channels per run in the first three datasets, 2 channels per run in the remaining ones.

How is the stabilisation done for physics data for detectors with unstable heater?

The alternative approach based on 2615 keV gamma rays as monoenergetic events is applied. In this case, the position of the ^{208}Tl peak is used as a reference to infer the evolution of signal event amplitudes with time and a second degree polynomial fit is performed to extract the stabilisation coefficients.

- table 1 is missing the Te-130 exposure
Thank you, we added this.

- 1328 the signal shape is described to come from calibration with 3 gaussians but it would be good to have in the main text the background shape that goes into the fit

The background components included in this fit are described in detail in the referenced Ph.D. theses, and for the sake of brevity we omit their description in the text here. The primary additional components are a flat background, a Compton shelf, and x-ray escape and coincidence peaks.

- how does the signal line shape influence the fit results and the limit?

The form of our signal lineshape is selected as the best fit to the calibration 2615 keV line, other lineshape functions should yield worse fit results. We expect a different lineshape form to have two primary possible effects. The first is a possible variation in the energy resolution, which we effectively already account for through our lineshape-resolution scaling procedure and the associated systematic uncertainties. The second potential consequence of selecting a

different lineshape function is to affect the tails of the ^{60}Co and $0\nu\beta\beta$ peaks in the fit. Since the latter, if observed, is very small, we expect changes to the tails of the peak lineshape to have negligible effects on the fit result. Regarding the ^{60}Co sum peak, since we already exclude channels with a FWHM above 19 keV from the data, we expect changes to the tails to have small effects on the $0\nu\beta\beta$ rate too.

- is the linear background shape supported by the background model?

Our background model currently shows weak evidence in favor of a linear slope in the background shape, which is why we permitted it in this fit as well. The fit results here also show a weak linear slope that is consistent with 0 at the 1 sigma level.

- I739 Stabilization is not clear, and why is it necessary here?

In this case the stabilization function derived from real data is necessary in order to inject signals resembling real events. When analyzing real events we reconstruct the energy of pulses by (i) raw amplitude evaluation, (ii) amplitude stabilization and (iii) applying the calibration function to stabilized amplitude. To map energy to raw amplitude for fake events we do exactly the opposite: given an energy, we invert the calibration function to deduce the corresponding stabilized amplitude, we then deduce the corresponding raw amplitude by inverting the stabilization function.

- I792 Can you comment on the dependence of the fit result on the choice of priors?

We employed a Gaussian prior probability distribution for all the systematics except the ones related to the lineshape scaling with energy for which we considered a multivariate prior extracted from the lineshape fits on gamma peaks in physics data. For the minimal model fit parameters, namely the signal rate, the ^{60}Co sum peak amplitude and the linear background coefficients we opted for a non-informative uniform prior. The reason behind this choice is mainly a conservative approach: we did not want to rely upon any prior assumption/knowledge on the possible neutrinoless double beta decay ($0\nu\beta\beta$) signal distribution as well as background components. We are aware that this choice is arbitrary and another possibility was a uniform prior on $\log(\Gamma)$, Γ being the signal rate, however since the $0\nu\beta\beta$ rate is proportional to the number of $0\nu\beta\beta$ events, which is something we directly measure with our experiments, we decided to set a flat prior on this parameter. We attempted to improve the discussion around the impact of alternative priors in the Methods section.

- I838 Do you include the Bayesian priors into the frequentist fit? the description of the frequentist analysis should have more details or at least a reference to the Rolke method

Yes, the frequentist analysis uses the same fit results as the Bayesian analysis, but simply performs a profile likelihood analysis of the scanned parameter space instead. Our Bayesian fit calculates the likelihood of every tested parameter configuration, which includes the

effects of the priors, and these likelihoods are what we use to obtain the profile likelihood of the $0\nu\beta\beta$ rate. Since we only use non-flat priors on the systematic uncertainties, that is the only area where the priors affect the Frequentist result. We have added a reference for the Rolke method and elaborated a little more on the Frequentist approach in the Methods section.

Language and style

- general: plural and singular are incorrect in some places
Thank you, we believe we have corrected this.

- abstract: inconsistent universe vs Universe
Thank you, we corrected this.

- l19 for non-physicists neutrino families and flavors might not be clearly the same thing
We corrected this in the re-worked text.

- it is a matter of taste but somehow the historical introduction reads like a bunch of statements without much connection

Thank you for this helpful perspective. We attempted to improve the focus of the introduction, limiting it to the relevance of Majorana neutrinos to the “big picture” of understanding why neutrinos are so light and testing B-L symmetry.

- l37 neutrino capable sounds like it can decide about its nature
We removed this in the re-worked text.

- l44 very strong statement "only realistic", there are efforts in HEP just more model dependent

This is a fair point, we have softened this statement in the re-worked text.

- fig 1 colors not visible in black and white

We updated the figure in question to improve the color contrast when printing in black and white.

- l186 how many datasets are there in the end? found only in table 1
We included this information within the text as well.

- l284 response function is a bit unusual formulation for the gamma line shape

We rely on the 2615 keV gamma line to extract the response to a monochromatic peak, this means our detector response function is indeed a gamma response function, but in any case we use it to infer how the crystals react to any possible energy deposition.

- 1734 add +for+ any physics run  run not defined, is it dataset?

A run represents a single unit of data acquisition and has a duration of approximately 24 hours. We group runs continuously acquired for one/two months into datasets. We added a brief sentence to specify it in the Data Analysis and Results section.

- 1738 this description of the trigger efficiency procedure should be before the discussion on advantages and disadvantages because now it is hard to read
Thank you, we reorganized the text as suggested to improve the clarity.

- 1776 the sentence has no
end
We fixed this.

In summary, I find that this paper after a major editorial revision should be published, but it would better fit in a physics journal, than Nature because of lack of interest from other fields.

Thank you, we appreciate your constructive comments and judgement on the physics-quality of our results. We do feel however that our article will be of interest to the wider community and we have tried to rework the article to give more prominence to the features of broader appeal.

In our opinion, the significance of our article to the wider community is the fact that we have demonstrated for the first time that a payload of approximately 1500 kg can be operated at 10mK continuously for multiple years. This has never been done before to our knowledge, not even in an industrial or military setting. During the first two years of CUORE data taking we had significant problems in operating continuously as illustrated in Fig. 2 of the updated article and prior to CUORE the coldest experiment with similar target mass was the Minigrail prototype which succeeded in operating for about 1 month but at a much higher base temperature of 65 mK. We have since solved these problems and with the 3rd and 4th year of operation demonstrated stable continuous data taking. 984 out of the 988 detectors are operating, demonstrating also that it is possible to build large complex arrays of these delicate detectors with high yield. Having reached this technological milestone, it now makes sense scientifically to optimize other aspects of the detector target for next generation $0\nu\beta\beta$ searches, namely enriching the target in the $0\nu\beta\beta$ isotope and adding active background suppression techniques, which is what the CUPID and AMORE projects plan to do. When we first proposed CUORE, it could not be taken for granted that cryogenic technology could meet the challenge.

In recent years, Nature has shown interest in publishing results from $0\nu\beta\beta$ detector experiments that have reached important technological milestones for the field. For example, Nature **544** 47–52 (2017) (GERDA) and Nature **510**, 229–234 (2014) (EXO-200). In a narrow sense these results could have been considered ‘updates’ relative to prior phases of

these experiments. However in the broader sense they were of interest to the wider community as they demonstrated these detector technologies were viable to pursue for next generation more sensitive experiments. We believe our new result, benefitting from our ability to collect large exposures is similarly of interest.

Accumulating large exposure has historically been a challenge for ultra low temperature cryogenic particle detectors searching for dark matter, like CRESST and CDMS. We believe the fact we have now demonstrated that cryogenic infrastructure can meet the requirement of large payloads (several hundred kg) and multiple years of continuous operation at base temperature is important for the future science reach that can be imagined with these types of detectors.

On the quantum information side, entities like IBM-Q are realizing that commercially available dilution refrigerators will be a limitation for their program of realizing $>10^6$ qubit quantum processors. Commercial vendors are naturally risk averse, we believe that broadly publicizing the challenges faced and overcome by CUORE is important to stimulate further advances in commercially available systems. Nature has recently published articles detailing the possible limitations of ionizing radiation on the stability of quantum processors, thus we feel our article highlighting the possibility of achieving simultaneously large capacity (payload and cooling power) but low-radioactivity dilution refrigerator systems will have resonance with the Nature readership who have been following the aforementioned articles.

Finally, as recently as July 2021, the field of neutrinoless double beta decay was the subject of a very appealing Nature news article, linked below. We submit that this indicates the journal considers important progress in this subfield to be of interest to its broad readership. doi: <https://doi.org/10.1038/d41586-021-01955-3>

Referee #2 (Remarks to the Author):

A. Summary of the key results

This paper is an update from CUORE's 2020 PRL with ~triple the exposure, reaching 1 t-y of exposure of TeO₂. Their neutrinoless double-beta decay limit comes out similar to the previous result but the sensitivity is a ~60% improvement.

B. Originality and significance: if not novel, please include reference

This is the article's weak point in my opinion. The CUORE technique is definitely original / novel compared to what came before it, and their limit is the best for ¹³⁰Te. However, this is not CUORE's first publication, and its originality and significance is weak especially compared to their article in Physics Review Letters just last year. Nature is usually in the

business of publishing ground-breaking results, however to me this seems more like a minor data update. The scientific effort involved in this specific paper appears to have essentially involved simply continuing to take data, and turning the analysis crank with a few minor improvements.

Even the milestone the authors claim to have achieved -- a 1 ton-year exposure -- requires some hand waving. To get to 1 ton-year, CUORE counts the full mass of deployed TeO₂ crystals. This includes both the double-beta decay isotope ¹³⁰Te and its "carrier materials" including oxygen. By this accounting, the CUORE exposure is far exceeded by other neutrinoless double-beta decay experiments, notably KamLAND-Zen with a 19 t-y Xe-LS exposure published in 2016. Of course what matters for the $0\nu\beta\beta$ result is just the mass in the active isotope, ¹³⁰Te. This exposure is only 1/3 of a ton year, which is also exceeded for example again by the KamLAND-Zen 2016 result, which achieved a 0.5 t-y isotopic exposure.

And CUORE is hardly the first ton-scale "cryogenic particle physics" experiment. It's not even the first one in LNGS, with the XENON1T WIMP Dark Matter detector running right down the hall from them. XENON1T is not nearly as cold as CUORE, still just last year they published a result that by CUORE accounting was a 2-ton-year exposure. It only falls below 1 ton year when you account for fiducialization, the role of which in this paper is played by the 88% containment efficiency, which would also bring CUORE below the 1 t-y threshold highlighted in the article's title.

For sure there is still a significant technical achievement here. I think the authors are trying to express that CUORE is perhaps the most technically challenging experiment of its kind performed to date (large-mass calorimetric experiment operated at sub-liquid-He temperatures), and 1 ton-year of exposure is indeed an achievement, even if it's an arbitrary goal post. However there are of course many most-challenging-of-its-kind-ever experiments, modern examples that come to mind are ATLAS/CMS, or within neutrino physics, for example KATRIN. In the end, one can always put enough qualifiers in a sentence to claim to have been the first to do something. The question is whether it's important. And even here the article spends almost no time arguing how CUORE's technical achievement is important. It contains essentially no broader or historical context for how or why 1 t-y of TeO₂ exposure is so much more important than their own 300 t-y last year, or 19 t-y or 2 t-y in other experiments. There is only a sentence or two about relevance to future $0\nu\beta\beta$ experiments and an extremely vague statement name-dropping quantum computing with no explanation or further discussion.

I'm a big fan of $0\nu\beta\beta$ physics and like to see it getting headlines in high-profile journals. And I do congratulate the CUORE team on their excellent result. However I have to admit that ultimately I'm surprised that this specific result is being considered for publication in Nature.

We thank the referee for this important perspective and appreciate the referee's enthusiasm

for $0\nu\beta\beta$ physics and our experiment! It is true that in one sense, our result can be viewed as merely an incremental improvement, and we agree that given the loose meaning of the word ‘cryogenic’ the goal of achieving a ton-yr exposure of a cryogenic target can seem arbitrary, or even unremarkable. Indeed the liquified natural gas industry certainly dwarfs the XENON1T experiment, CUORE, and many others in terms of cryogenic payload.

The benchmark of a tonne-scale payload is one that has some resonance in the $0\nu\beta\beta$ community as it is laid down in the US 2015 Long Range for Nuclear Science as a target for next-generation $0\nu\beta\beta$ searches. We agree that we may have taken the significance of this to the wider scientific community for granted.

Thus we have tried to improve the article in this regard. In our opinion, the significance of our article to the wider community is the fact that we have demonstrated for the first time that a 1500 kg payload can be operated at 10 mK continuously for multiple years. This has never been done before to our knowledge, not even in industrial or military settings where one is usually not burdened with the constraints that come with extracting physics from the target. During the first two years of CUORE data taking we had significant problems in operating continuously as illustrated in Fig. 2 of the updated article and prior to CUORE the coldest experiment with similar target mass was the Minigrail prototype which succeeded in operating for about 1 month but at a much higher base temperature of 65 mK. We have since solved these problems and with the 3rd and 4th year of operation demonstrated stable continuous data taking. Moreover 984 out of the 988 detectors are operating, demonstrating that it is possible to build large complex arrays of these delicate detectors with high yield. Having reached this technological milestone, it now makes sense scientifically to optimize other aspects of the detector target for next generation $0\nu\beta\beta$ searches, namely enriching the target in the $0\nu\beta\beta$ isotope and adding active background suppression techniques, which is what the CUPID and AMORE projects plan to do.

In recent years, Nature has shown interest in publishing results from $0\nu\beta\beta$ detector experiments that have reached important technological milestones for the field. For example, Nature **544** 47–52 (2017) (GERDA) and Nature **510**, 229–234 (2014) (EXO-200). In a narrow sense these results could have been considered straightforward improvements over prior phases of these experiments. However in the broader sense they were of interest to the wider community as they demonstrated these detector technologies were viable to pursue for next generation more sensitive experiments. We believe our new result, benefitting from our ability to collect large exposures is similarly of interest.

Due to the challenges of scaling cryogenic infrastructure, it has long been a daunting proposal that calorimeters could compete with other $0\nu\beta\beta$ technologies at the tonne scale. We demonstrate that not only are we able to instrument at this scale, but that we can operate stably, efficiently, and with excellent sensitivity over years of data collection. We demonstrate that cryogenic (~ 10 mK) calorimeters are very much on the forefront of the search for $0\nu\beta\beta$.

Accumulating large exposure has historically been a challenge for ultra low temperature cryogenic particle detectors searching for dark matter, like CRESST and CDMS. We believe the fact we have now demonstrated that cryogenic infrastructure can meet the requirement of large payloads (several hundred kg) and multiple years of continuous operation at base temperature is important for the future science reach that can be imagined with these types of detectors.

On the quantum information side, entities like IBM-Q are realizing that commercially available dilution refrigerators will be a limitation for their program of realizing $>10^6$ qubit quantum processors. Commercial vendors are naturally risk averse, we believe that broadly publicizing the risks faced and overcome by CUORE is important to stimulate further advances in commercially available systems. On another front, Nature has recently published articles detailing the possible limitations of ionizing radiation on the stability of quantum processors, thus we feel our article highlighting the possibility of achieving simultaneously large capacity but low-radioactivity dilution refrigerator systems will have resonance with the Nature readership who have been following the aforementioned articles.

Finally, as recently as July 2021, the field of neutrinoless double beta decay was the subject of a very appealing Nature news article, linked below. We submit that this indicates the journal considers important progress in this subfield to be of interest to its broad readership. *doi: <https://doi.org/10.1038/d41586-021-01955-3>*

C. Data & methodology: validity of approach, quality of data, quality of presentation

- Line 212 and 303: Why does the energy scale require a 2nd order polynomial fit? What is the source of the nonlinearity?

L212: the 2nd order fit is because the amplitudes of the peaks studied do not scale linearly with the energy deposit, most likely as a result of the non-linear behavior of the NTD thermistors we use.

L313: To evaluate possible systematic shifts in the position of gamma peaks in physics data we perform a fit on known CUORE contaminations to account for both an energy dependence and possible discrepancy in the detector response function between calibration and physics data. The residuals, defined as the difference between the literature value of the peak energies and the reconstructed ones, exhibit a parabolic energy dependence which we attribute to possible systematic effects of how our calibration peaks are spaced out in energy. We take them into account by shifting the expected signal peak position on a channel basis.

- It was unclear to me how precisely the salting method works. Is it applied at the level of rawdata? That seems difficult since the raw data is usually integer in nature, so scaling by a factor of 2527 / 2615 would make things obviously non-integer and detectable by analysts. Conversely it could be done at the energy reconstruction stage, but then it seems one could simply re-analyze the raw data and conclude whether an event was "salt" or not. Some better explanation would help.

The salting procedure to blind our data in the region of interest is applied after the energy reconstruction stage and the coincidences identification, but before the pulse shape analysis and the efficiency evaluation. We shift the reconstructed energy of a random fraction of events from the 2615 keV ^{208}Tl peak down by 87 keV towards the Q-value and the same fraction of events from 2527 keV to the thallium line. All the subsequent analysis steps, including the choice of the fit model and parameters, are performed on blinded data affected by this shift in the spectrum. The salting procedure is reversed only once everything is fixed.

It is true that an analyst could see through the salting if they really tried, but the goal of our blinding procedure is to prevent ourselves from making biased decisions on how to tune our cuts and fitting procedure, not to block someone who is actively trying to break through. On top of that, we consider low level data processing (such as stabilization, energy calibration) and high level analysis for the $0\nu\beta\beta$ search (e.g. efficiency evaluation and final spectrum fit) as two distinct steps of our analysis and different groups of people are responsible for them. Furthermore, standard checks on the status of data processing steps include minimal tests to verify the success of the applied algorithms rather than analysis of the physical quantities extracted from them.

- The PSD method described in lines 236-240 is claimed to remove pileup events and non-physical pulses, yet the Figure 2 lower-right panel shows it to also affect the calibration peak width. I suspect the PSD is also being used to remove "noisy waveforms" as is done in the 2020 PRL. If this is the case, the description should be updated to reflect this.

It is the case that PSD also removes waveforms that have excessive noise or other kinds of unstable conditions (such as a baseline that hasn't yet fully relaxed since the previous event), which partially accounts for the improvement in calibration resolution. The description has been updated accordingly.

- On the same topic -- the PSD does not seem to also sharpen the peaks in the physics data shown in the lower-left plot of Fig 2. Is this a problem for the resolution function model? Some comment should be added on this. See also my lengthy comments on the resolution model below.

Most of the improvement in calibration resolution from the PSD is the result of removing pileup events or events where the baseline has not yet fully relaxed (which is a kind of pileup,

but the previous pulse would not be visible in the event window), which would otherwise widen the peak. These kinds of events rarely occur in physics data, where the event rate is much lower, so we do not think it is unusual for there to be little improvement in the physics peak resolution from the PSD.

- The selection criteria paragraph starting on line 223 does not describe the "basic quality cuts" mentioned in line 257. What are these? Their description should be added somewhere.

With "basic quality cuts" we refer to pulse quality requirements based on raw pulse estimators, which we impose in order to have a single pulse-like feature in the event window and a stable pre-trigger voltage. We added a sentence to specify it.

- The last sentence of p3 states that the reconstruction efficiency is computed by-calorimeter and then averaged over the entire dataset. Why is this necessary? Is it highly variable? If so, why?

The reconstruction efficiency is very similar among calorimeters, but some calorimeters do not have functional heaters and so cannot have this efficiency directly calculated. We average over the entire dataset to cover the heaterless calorimeters, as well as for computational efficiency purposes in the final 0v fit. This efficiency with its uncertainty is included in the Bayesian fit as a Gaussian prior, and it would be computationally-prohibitive to treat all 900+ efficiencies per dataset as separate systematics in the fit.

- Fig 2, lower-left panel: the PSD removes much more continuum (factor of ~ 2) than peak. Why is this the case? It's hard to believe that it's due to pileup, the effect of which is already only a factor of ~ 2 reduction in the vastly higher-rate calibration data shown in the lower-right plot. Some comment on this should be added.

In the physics data, the PSD removes mostly noise instead of pileup. This noise tends to be the result of mechanical perturbations to the cryostat or other transient instabilities in the operating conditions, which can sometimes emulate the shape of a normal thermal pulse and thus pass our previous basic data quality cuts. We try to eliminate these kinds of events when we eliminate data from time periods with poor conditions, but we can't catch them all with that manual procedure, and so the ones that make it through show up throughout the continuum and end up cut by the PSD instead. We added a comment on this in Methods.

- In the paragraph starting on line 271, it's described that physics data is used to extract the PSD efficiency, and statistics are too low to allow for a by-calorimeter measurement. Why can't the PSD efficiency be extracted directly from high-stats calibration data?

Our PSD efficiency calculation for a peak relies on being able to perform a background

subtraction, so that we can determine how many “signal” events were in the peak before and after the cut. In physics data this can be done with the flat background + Gaussian signal that we fit. In calibration the background shape is more complex and pileup is much higher and dominates the PSD efficiency there. As a result, the number derived from calibration data does not give us a reliable estimate of the efficiency in physics data.

- I was particularly concerned about the detector response handling. Does the same triple-gaussian parameterization used for the 2615 calibration peak work for all peaks across all energies? The references only characterize FWHM(E) but I could not tell whether e.g. the parameters of each gaussian are varying in a sensible way across E. More importantly, it seems perfectly likely that the non-gaussian behavior is associated with the position distribution of energy depositions in the crystals. $0\nu\beta\beta$ is very different in this respect from 2615 keV gamma rays, so I worry that the uncertainty in the $0\nu\beta\beta$ peak shape is underestimated, which would impact the final extracted limits.

The resolution function parameterization is not just important for the $0\nu\beta\beta$ peak, but also for the ^{60}Co sum peak where there are 2 gammas and probably a much broader spatial distribution. This one is so important because its proximity to $Q_{\beta\beta}$ (anti)correlates the peak width with the extracted $0\nu\beta\beta$ rate. So the resolution function comes into the final result effectively twice, for two peaks with fundamentally different energy deposition topologies from each other and from the peak used to characterize the empirical resolution function. It's thus critical to have some demonstration on whether or not the empirical model works for all event topologies.

In addition to checking ^{60}Co (in calibration and physics data), I recommend also doing the same test on other peaks from interactions with fundamentally different topologies. For example, double-escape peak events in the calibration data have a much more localized energy deposition, although positron annihilation-in-flight leads to some extra width that can be estimated with MC. The single escape peak provides another good test case. Low-energy peaks are also more single-site and more localized near detector surfaces and would provide another important test case.

We completely agree that an extensive analysis of the detector response as a function of energy and event topology is mandatory and in fact we studied this. We tested the monochromatic peak shape extracted on calibration data on several gamma and beta peaks with different topologies and in a wide energy range from 500 keV to 2615 keV on physics data. Among them, we evaluated the shape of signal events on the 1173, 1332 keV gamma peaks from ^{60}Co and 1460 keV from ^{40}K , all of them are single particle peaks and as such single crystal events.

We also considered the 510.7, 583 and 2615 keV peaks from ^{208}Tl , whose topology is different because of the simultaneous emission in the beta decay of more than one gamma ray. At a later stage, we discarded the 510.7 keV peak due to the unavoidable broadening caused by the partial overlap with the 511 keV peak from e^+e^- annihilation. For all the listed

peaks we performed the line shape fit leaving the peak position and resolution free to float in order to extract for both of them the correct parameterization at the posited signal peak Q_{bb} and the ^{60}Co sum peak. Checking the shape on the double-escape and single-escape peaks is a good idea, but we lack the statistics to get a reliable estimate of the energy bias and resolution scaling on those peaks in physics data, and we do not calculate this bias and scaling for calibration data. To check the variation in the 3-Gaussian shape, we would have to do this on a detector-by-detector basis, and even referring to calibration data we do not have sufficient statistics on those peaks to do this. In our current analysis, we just note that with the limited statistics we have, the escape peaks do not show significant differences in terms of bias and resolution compared to the other physics peaks we use to quantify these effects, and so any minor differences should be accounted for by the systematic uncertainty we impose on the detector response scaling.

- It was also concerning to me that in lines 305-306 the manuscript described the variation of FWHM with energy to be linear, while the PhD theses referenced in the article both appear to show non-linear trends. Some comment or demonstration of linearity should be added.

We changed our approach to fit the FWHM as a function of energy based on an accurate study of the resolution trend in all the datasets included. We found that a linear description of the energy resolution as a function of energy provided a zero keV FWHM consistent with the one extracted from noise events (no signal peak in the window). However, comparing the pol-2 and pol-1 fits we found that the change in the fit function has negligible effects at the Q-value since non linear effects were mostly induced by the reconstructed 2615 keV TI line position. The details on this scaling have been left out in the new version of the paper due to space constraints.

D. Appropriate use of statistics and treatment of uncertainties

The Bayesian limit uses a flat prior on the $0\nu\beta\beta$ rate with a physical boundary imposed to exclude negative rates. Such limits are notoriously prior-dependent. The sensitivity to choice of prior for the $0\nu\beta\beta$ rate should be evaluated and reported (in the Methods section).

We have added a paragraph in the Methods section mentioning 3 other choices we could have as uninformative priors (uniform on mbb, scale invariance by being uniform on $\log(\text{rate})$, and uniform on half-life).

The frequentist limit description is missing a reporting of the median sensitivity, this should be added to the last paragraph in the Methods section. It would also be good to report the frequentist χ^2 in Fig. 3 (for example with a separate right-axis scale).

Thank you for this suggestion. We have added the median Frequentist sensitivity, and have added a plot of the frequentist delta chi2 alongside the Bayesian posterior pdf in the Extended Data.

E. Conclusions: robustness, validity, reliability

The Discussion section points out that the limit is weaker than in the 2020 PRL. To aid in this explanation, the authors might also point out that the previous limit was 2x better than its sensitivity -- well within expected fluctuations, but made it more likely that the next result (this limit) would be weaker.

Thank you for this suggestion. We added a couple sentences on this topic in the Discussion section

F. Suggested improvements: experiments, data for possible revision

I recommend further study of the energy resolution function, see comments in (C)

G. References: appropriate credit to previous work?

- A reference is missing for the data salting method where one can find details on how it is done.

We added a reference to clarify the blinding procedure employed.

- A reference is missing for the Rolke method mentioned in line 837. Added reference.

H. Clarity and context: lucidity of abstract/summary, appropriateness of abstract, introduction and conclusions

It's a bit strange that the abstract has 5-6 lines about leptogenesis, while the intro only mentions it in the last sentence of the first paragraph after a long exposition on the history of neutrinos. In my opinion, the intro should be re-written to match the much more attention-grabbing abstract.

Thank you for this suggestion. We re-wrote the introduction to improve the focus.

I also noticed a couple of instances of redundancy, which was surprising for such a short article:

- The containment efficiency is described twice, once near line 231 and again near line 253.

- The quadratic form of the energy scale was mentioned twice, once in 212 and again in 303, without reference to the earlier text. This might confuse the reader into thinking that they are two different things when as far as I can tell they are the same.

Regarding containment efficiency, we eliminated the repetition.

Instead, in the lines you mention as describing both second degree fits, we are actually referring to different steps of the analysis. The former (line 212) describes the calibration procedure we employ in the data processing chain, which indeed features a second order polynomial fit of ^{232}Th and ^{60}Co peaks amplitudes to extract calibration coefficients. The latter (line 303) describes the way we model the energy dependence of the peak position in physics data to infer the position where to expect a $0\nu\beta\beta$ signal. We hope this clarifies the difference.

Referee #3 (Remarks to the Author):

The authors present a comprehensive analysis from the first tonne-year exposure of CUORE, searching for $0\nu\beta\beta$ decay of Te-130. While the half-life limit is in line with their previous result (albeit increased statistics), this is the world's most sensitive search for $0\nu\beta\beta$ decay with Te-130 and a technological milestone that paves the way for a much improved upgrade, called CUPID. I believe this technological achievement, along with an improved analysis method, merits publication in Nature.

Overall, the manuscript is well-written, well-organized, and a pleasure to read. However, I do have several comments that I hope the authors consider before the manuscript is accepted for publishing. They are as follows:

1) Lines 63-64: "...anti-coincidence efficiency is the probability that a signal event is not incorrectly vetoed due to an accidental coincidence between two independent events." Did you really mean to use a double negative here? Should it not be "correctly vetoed" instead of "not incorrectly vetoed"?

Yes, we did. However, we can see that this was a poor choice of phrasing. We changed the wording to remove the double negative. Thank you for the suggestion.

2) Extended Data Figure 8: The distribution of pseudo experiments shows the familiar shape we typically see in these kinds of results. However, when I follow the diminishing tail of the distribution towards longer half-life, I notice that the very last bin near 10^{26} years contains about 20 experiments. It's noticeably higher than all the preceding bins and almost makes the reader wonder if something strange is happening in the background-only fitting procedure. Of course, this small number of pseudo experiments will only have a very small effect on the median limit, but it would be good to confirm that this strange bin is not a

symptom of some underlying problems with the fit.

After inspecting the results more closely, we found this was the result of an error in the binning procedure that caused the toys with results $>10^{26}$ years to all end up in the last bin. Thanks for catching this - the plot has been updated appropriately.

3) Extended Data Figure 7: The table listing the various systematics has a column describing the different priors used. I notice that "Analysis efficiency II" referring to the PSD efficiency assumes a Gaussian prior. This is an important difference from the previous analysis reported in DOI: 10.1103/PhysRevLett.124.122501, where a uniform prior was assumed. I'd prefer to see a sentence or two in the body of the text explicitly describing this difference, since many readers will likely be comparing the present analysis with the previous one. Why wasn't this available as a Gaussian prior in the previous analysis?

In the previous analysis, we introduced a global uncertainty on the PSD efficiency because there was a discrepancy in the efficiencies obtained from looking at one-crystal versus two-crystal events. Because we did not understand the source of this discrepancy, we used a uniform prior to account for this uncertainty. In the current analysis, we performed the same study and were able to understand the source of the difference - we confirmed the existence of non-negligible amounts of two-crystal events that are the result of false coincidences, which make them unsuitable for PSD efficiency evaluation.

In this analysis, the PSD efficiency systematic uncertainty is the result of possible variations in efficiency between individual detectors, and this is accounted for by the method described in the text. However, this time we believe that our method of evaluating the nominal value of the PSD efficiency as relevant to the $0\nu\beta\beta$ analysis is correct (using the exposure-weighting), so we use a Gaussian prior instead.

4) Discussion Section: As the authors state in the Discussion section, the analysis presented in this manuscript is weaker than their previous result and is well within the expected range of outcomes due to statistical fluctuations. While statistical fluctuations are out of our control, the data analysis seems quite mature at this point and has the added improvement of PCA used to reject background with higher signal selection efficiency. It would be beneficial to highlight the maturity of the CUORE analysis at this stage in the experiment and provide readers with a statement concerning any further improvements in the analysis (absent the major hardware upgrade for CUPID). Is this CUORE analysis as good as it gets, and is the experiment now limited by backgrounds in the ROI?

The CUORE analysis is indeed mature at this point and CUPID will rely on tools and procedures we validated and optimized through these years. However, we are still working on two sides to further improve it. On one hand, an extensive analysis is currently ongoing to understand, disentangle and possibly reduce the noise sources that affect our energy

resolution at the Q-value in order to bring it to our target value.

On the other hand, we are trying to optimize the coincidence analysis exploiting the information coming from other interesting physics analyses either completed, as is the case of the $0\nu\beta\beta$ search to the first 0^+ excited state of ^{130}Xe , published on EPJC this year and available at the link <https://link.springer.com/article/10.1140/epjc/s10052-021-09317-z>, or ongoing, such as the study of ^{128}Te $0\nu\beta\beta$ decay and ^{120}Te $0\nu\beta\text{EC}$ decay. This could have a strong impact on our comprehension of CUORE contaminations and thus lead to improvements in our background model in view of CUPID as well as imply a further reduction of our background index in the region of interest.

Reviewer Reports on the First Revision:

Referee #1:

I thank the authors for the careful consideration of all my comments and suggestions. The paper has been largely rewritten. The modifications increased the quality and I am happy to suggest it for publication.

Some small remarks for your consideration:

- line 28 "low-energy experimentally accessible process" is hard to read
- line 38 add Collaboration after CUORE
- line 81 not clear where is the polyethylene and boric acid layer
- fig. 2 without a comment on the further data taking additional to the data in this paper, I would not show the exposure collection of the last year
- line 121 forward reference to fig. 4 is not very elegant
- line 124 background index is not defined and might be hard to understand at this point in the paper
- line 204 background is made -> add "of"
- line 230 competitiveness -> add "of"
- line 233 the symbol 1 tonne_iso yr could be introduced already in tab. 1 so one can recognize which numbers to compare
- line 660 "is not vetoed due to" is hard to understand
- Some figures in the appendix are referenced as Extended Data Fig. but not all e.g. line 643
- Two figures are not references anywhere in the paper: Fig. 5 and Fig. 7

Referee #2:

This version of the manuscript represents quite a significant revision.

I find that the relevance of the work for publication in Nature is much better argued in this revision, especially in the abstract and in the Impact section. The case is still, in my opinion, quite a bit weaker than in the previous two Nature articles on $0\nu\beta\beta$ from EXO-200 and GERDA, both of which represented world-leading sensitivity in mbb at the time they were released. The leading limits are now held by KamLAND-Zen and GERDA, and this result does not exceed those, and only marginally improves the limit for ^{130}Te over the existing limit from CUORE published just last year. So as it is, the argument for publishing nature is mostly one about instrumentation and technique. If this is sufficient for the Nature editors then I am happy to defer to their judgement. However:

1) I think it is still important for publication in Nature that mention be made -somewhere- that this represents a world-leading sensitivity at least in ^{130}Te . I was a bit shocked to find that all references to CUORE's 2020 PRL have been completely removed in trying to shorten the article,

save one comment about triggering in the Methods section (despite the author's reply to section "E" of my comments that this discussion had been updated). This work must be put in the context of CUORE's most recent published results. Also desired is a short paragraph in the Methods section highlighting any analysis changes relative to the 2020 PRL. For example, the updated choice of PSD prior noticed by Referee 3 should be mentioned there.

Beyond that, in general I found that the vast majority of my comments were sufficiently addressed in the revised manuscript. I only have two other major remaining concerns:

2) The Fig 7 caption was updated to explain that the PSD cut "mostly eliminates random noise events that emulate the shape of physical pulses." As written this is in conflict with the description in lines 176-179, which says that the PSD removes:

- pulses consistent with more than one energy deposit (not noise)
- pulses with a non-physical shape (do not "emulate the shape of a physical pulse")
- excessively noisy pulses (physics pulses with noise on top of them)

If the "random noise events" are one of these three categories, the terminology should be made consistent. If they are an additional type of event, their description should be added to the paper.

Either way, from Figs 4 and 7 this background component seems to be the dominant background before cuts, so its potential residual contamination after cuts needs to be quantified. This is important because in the paragraph starting on line 200 it says that the continuum after cuts is $\sim 90\%$ alphas and $\sim 10\%$ Compton scatters. This implies that any residual background from random noise events is negligible, but that is not demonstrated in this manuscript, nor could I find it in references 31 and 33 that appear next to these statements. While a full characterization of the background is not necessary for $0\nu\beta\beta$ analysis presented in this paper, the conclusion that the background is essentially all alphas and gammas is relevant for future similar experiments based on the CUORE technique like CUPID.

If an upper limit on the residual "random noise event" background is extracted rigorously in another publication, that should be cited near line 205. However if this high random noise background is not described elsewhere, its possible residual contamination after applying the PSD cuts should be discussed before claiming that the remnant backgrounds are $\sim 100\%$ physics events.

3) This comment is in reference to the last 2 bullets in section "C" of my comments on the original draft. I was satisfied with the authors' response to my questions, however it appears that not only were those responses not folded into the text, some of the original essential details were removed. These details are essential because the effective background and thus the sensitivity of the experiment scales directly with the peak width, and the empirical triple-gaussian used by CUORE is not standard enough to hand-wave with the qualitative text of lines 183-198 couched in terms of a single FWHM. That text only cites reference 32, which is from 2018, prior to the studies described in the authors' responses to my comments, and so is insufficient to justify the treatment here. If a publication exists on the studies described in the response to my comments by the authors to justify the empirical peak shape model that is more recent than the PhD theses that still treated the energy variation as non-linear, then it would be sufficient to just reference that here. If such a publication does not exist, the arguments and studies described in the authors' responses to my comments should be incorporated at least briefly in the Methods section.

I have one other minor technical issue to raise:

(4) Since the background near Q_{bb} is sloped, it is not clear what is meant exactly by "background index." Line 720 implies that the "index" is independent from the "linear slope" in the background, but fails to indicate at what energy that index is referenced. I suspect it is Q_{bb} but that needs to be stated explicitly. More accurately, it is important that the background index represent the value of the total background (including, in principle, the residual ^{60}Co tail) either at Q_{bb} or averaged over some energy range like 1 FWHM. Anyway, please make the definition of background index precise and explicit somewhere.

Finally, a suggestion:

(5) I thank the authors for adding so much detail on the exploration of the prior dependence. Given the outcome, I think it would be sufficient to just state in a sentence or two that "uniform priors on m_{bb} , $T_{1/2}$, and $\log(\text{rate})$ all lead to more aggressive limits by 25%-x4, and also require a lower cutoff on Γ , set to 10^{-27} years (~ 1 signal event)." This could open up some room for other more essential details. If there is space for the full description, I believe the flat prior on m_{bb} is equivalent to the Jeffreys prior, that might be worth noting since it's a common alternative to the flat prior.

Referee #3:

The authors present a comprehensive analysis from the first tonne-year exposure of CUORE, searching for $0\nu\beta\beta$ decay of Te-130. While the half-life limit is in line with their previous result (albeit increased statistics), this is the world's most sensitive search for $0\nu\beta\beta$ decay with Te-130 and a technological milestone that paves the way for a much improved upgrade, called CUPID. I believe this technological achievement, along with an improved analysis method, merits publication in Nature.

All of my questions were sufficiently answered or corrected in the manuscript, including a minor binning error for the 90% CI limit plot, and many of the grammar mistakes. I would say that this most recent draft is ready for publication and I recommend doing so.

I'd like to commend the authors for answering the all the hard questions posed by the other referees, including the comments about whether this paper was novel or groundbreaking enough to merit publication in Nature. Indeed, as pointed out by the authors, there have been other similar papers by two competing experiments (GERDA and EXO-200) that were also published in Nature very recently.

Author Rebuttals to First Revision:

Questions and answers to referees

Referee #1 (Remarks to the Author):

I thank the authors for the careful consideration of all my comments and suggestions. The paper has been largely rewritten. The modifications increased the quality and I am happy to suggest it for publication.

Some small remarks for your consideration:

- line 28 "low-energy experimentally accessible process" is hard to read

We changed the sentence to "Experimental searches for $0\nu\beta\beta$ decay are the most sensitive means to corroborate this framework".

- line 38 add Collaboration after CUORE

We added this.

- line 81 not clear where is the polyethylene and boric acid layer

We changed the sentence to "Environmental neutrons are suppressed by a 20-cm layer of polyethylene and a thin layer of boric acid outside of the external lead shield."

- fig. 2 without a comment on the further data taking additional to the data in this paper, I would not show the exposure collection of the last year

We decided to leave it as it shows the continuity of CUORE data taking.

- line 121 forward reference to fig. 4 is not very elegant

Thanks, we removed the sentence entirely.

- line 124 background index is not defined and might be hard to understand at this point in the paper

We removed the sentence entirely along with the mention of the background index at this point, replacing them with "background rate".

- line 204 background is made -> add "of"

This sentence has now been reworded.

- line 230 competitiveness -> add "of"

We added this.

- line 233 the symbol 1 tonne_iso yr could be introduced already in tab. 1 so one can recognize which numbers to compare

Thanks, to avoid changing tab.1 we modified the line as follows: "designed ^{130}Te exposure of 1000 kg·yr".

- line 660 "is not vetoed due to" is hard to understand

We changed the sentence to "The AC efficiency is the probability that a true single-crystal event correctly passes our AC cut, instead of being wrongly vetoed due to an accidental coincidence with an unrelated event."

- Some figures in the appendix are referenced as Extended Data Fig. but not all e.g. line 643

Thanks, we fixed this.

- Two figures are not references anywhere in the paper: Fig. 5 and Fig. 7

We added references to these figures. We also changed the numbering of all figures in the Extended Data sections as Extended Data Fig. 1-5.

Referee #2 (Remarks to the Author):

This version of the manuscript represents quite a significant revision.

I find that the relevance of the work for publication in Nature is much better argued in this revision, especially in the abstract and in the Impact section. The case is still, in my opinion, quite a bit weaker than in the previous two Nature articles on $0\nu\beta\beta$ from EXO-200 and GERDA, both of which represented world-leading sensitivity in mbb at the time they were released. The leading limits are now held by KamLAND-Zen and GERDA, and this result does not exceed those, and only marginally improves the limit for ^{130}Te over the existing limit from CUORE published just last year. So as it is, the argument for publishing nature is

mostly one about instrumentation and technique. If this is sufficient for the Nature editors then I am happy to defer to their judgement.

However:

1) I think it is still important for publication in Nature that mention be made -somewhere- that this represents a world-leading sensitivity at least in ^{130}Te . I was a bit shocked to find that all references to CUORE's 2020 PRL have been completely removed in trying to shorten the article, save one comment about triggering in the Methods section (despite the author's reply to section "E" of my comments that this discussion had been updated). This work must be put in the context of CUORE's most recent published results. Also desired is a short paragraph in the Methods section highlighting any analysis changes relative to the 2020 PRL. For example, the updated choice of PSD prior noticed by Referee 3 should be mentioned there.

Our apologies for some inconsistency between our last response to the referee and some of the bigger edits we made to the draft. In order to better contextualize this result in comparison to our last one, we have added to lines 223-224 an explicit reference to the 2020 PRL result, with the mention that this result has the world-leading sensitivity in Te^{130} , and that the sensitivity has improved as expected since our previous result even though the limit is weaker. With regards to some of the other points on differences in analysis, we've made an addition detailed in the response to point (3), and we have also added a sentence in the Efficiencies section of Methods that discusses PSD systematic uncertainty, stating "This takes a Gaussian prior instead of the uniform prior used in our previous result, which had its uncertainty come from a discrepancy between two approaches that has since been resolved."

Beyond that, in general I found that the vast majority of my comments were sufficiently addressed in the revised manuscript. I only have two other major remaining concerns:

2) The Fig 7 caption was updated to explain that the PSD cut "mostly eliminates random noise events that emulate the shape of physical pulses." As written this is in conflict with the description in lines 176-179, which says that the PSD removes:

- pulses consistent with more than one energy deposit (not noise)
- pulses with a non-physical shape (do not "emulate the shape of a physical pulse")
- excessively noisy pulses (physics pulses with noise on top of them)

If the "random noise events" are one of these three categories, the terminology should be made consistent. If they are an additional type of event, their description should be added to the paper.

Either way, from Figs 4 and 7 this background component seems to be the dominant background before cuts, so its potential residual contamination after cuts needs to be quantified. This is important because in the paragraph starting on line 200 it says that the continuum after cuts is ~90% alphas and ~10% Compton scatters. This implies that any residual background from random noise events is negligible, but that is not demonstrated in this manuscript, nor could I find it in references 31 and 33 that appear next to these statements. While a full characterization of the background is not necessary for $0\nu\beta\beta$ analysis presented in this paper, the conclusion that the background is essentially all alphas and gammas is relevant for future similar experiments based on the CUORE technique like CUPID. If an upper limit on the residual "random noise event" background is extracted rigorously in another publication, that should be cited near line 205. However if this high random noise background is not described elsewhere, its possible residual contamination after applying the PSD cuts should be discussed before claiming that the remnant backgrounds are ~100% physics events.

The "random noise events" described in Fig 7 caption referred to both pulses that look like physical energy deposits but with excessive noise and pulses that have non-physical shapes and look to be due to other sorts of noise. The phrase "emulate the shape of a physical pulse" was meant to include this latter class as well, since those events still have the basic structure of a rise in temperature followed by a decay towards the baseline, allowing them to pass our trigger algorithm and other basic data quality cuts, but closer inspection (either by hand or by the PSD algorithm) shows they are indeed non-physical. The caption has been updated to be more specific about this.

For this analysis, we found no residual contamination of obvious noise events in the ROI after conducting a manual inspection of the pulses of all events in our final spectrum. We have reworded the sentence slightly as "We estimate ~90% of the continuum background consists of degraded alpha particles from radioactive contaminants of the support structure surface, while the other ~10% are multi-Compton scattered 2615 keV gamma events", which hopefully clarifies that this is our estimate from our referenced background models.

3) This comment is in reference to the last 2 bullets in section "C" of my comments on the original draft. I was satisfied with the authors' response to my questions, however it appears that not only were those responses not folded into the text, some of the original essential details were removed. These details are essential because the effective background and thus the sensitivity of the experiment scales directly with the peak width, and the empirical triple-gaussian used by CUORE is not standard enough to hand-wave with the qualitative text of lines 183-198 couched in terms of a single FWHM. That text only cites reference 32, which is from 2018, prior to the studies described in the authors' responses to my comments,

and so is insufficient to justify the treatment here. If a publication exists on the studies described in the response to my comments by the authors to justify the empirical peakshape model that is more recent than the PhD theses that still treated the energy variation as non-linear, then it would be sufficient to just reference that here. If such a publication does not exist, the arguments and studies described in the authors' responses to my comments should be incorporated at least briefly in the Methods section.

We have added a sentence at line 196 to elaborate on the scaling and how the resolution scaling function has been changed since our previous result: "The bias is allowed to scale as a quadratic function of energy as done in our previous result, while the resolution scaling has been changed to a linear function of energy, following studies showing that it was overparameterized by a quadratic scaling." The 3-Gaussian response function itself, though indeed still non-standard and not fully explained, is still unchanged in its basic form since our previous results, and so we leave that part to the references.

I have one other minor technical issue to raise:

(4) Since the background near Q_{bb} is sloped, it is not clear what is meant exactly by "background index." Line 720 implies that the "index" is independent from the "linear slope" in the background, but fails to indicate at what energy that index is referenced. I suspect it is Q_{bb} but that needs to be stated explicitly. More accurately, it is important that the background index represent the value of the total background (including, in principle, the residual ^{60}Co tail) either at Q_{bb} or averaged over some energy range like 1 FWHM. Anyway, please make the definition of background index precise and explicit somewhere.

The background index does indeed refer to the value at Q_{bb} - this has now been clarified in line 212. Given our typical energy resolutions, the ^{60}Co tail is too small to contribute to our background index at Q_{bb} at the significant figures we cite.

Finally, a suggestion:

(5) I thank the authors for adding so much detail on the exploration of the prior dependence.

Given the outcome, I think it would be sufficient to just state in a sentence or two that "uniform priors on m_{bb} , $T_{1/2}$, and $\log(\text{rate})$ all lead to more aggressive limits by 25%-x4, and also require a lower cutoff on Γ , set to 10^{-27} years (~ 1 signal event)." This could open up some room for other more essential details. If there is space for the full description, I believe the flat prior on m_{bb} is equivalent to the Jeffreys prior, that might be worth noting since it's a common alternative to the flat prior.

We have added a remark at line 755 that the flat prior on mbb is equivalent to using the Jeffreys prior. Since these details are part of the Methods section, we believe the space constraints shouldn't be an issue here, but we can cut this down as suggested if necessary.

Referee #3 (Remarks to the Author):

The authors present a comprehensive analysis from the first tonne-year exposure of CUORE, searching for $0\nu\beta\beta$ decay of Te-130. While the half-life limit is in line with their previous result (albeit increased statistics), this is the world's most sensitive search for $0\nu\beta\beta$ decay with Te-130 and a technological milestone that paves the way for a much improved upgrade, called CUPID. I believe this technological achievement, along with an improved analysis method, merits publication in Nature.

All of my questions were sufficiently answered or corrected in the manuscript, including a minor binning error for the 90% CI limit plot, and many of the grammar mistakes.

I would say that this most recent draft is ready for publication and I recommend doing so.

I'd like to commend the authors for answering the all the hard questions posed by the other referees, including the comments about whether this paper was novel or groundbreaking enough to merit publication in Nature. Indeed, as pointed out by the authors, there have been other similar papers by two competing experiments (GERDA and EXO-200) that were also published in Nature very recently.

Thanks for your comment.